# Retrieving $CH_4$ emission rate from coal mine ventilation shaft using UAV-based AirCore observations and the GA-IPPF model

Tianqi Shi[1], Zeyu Han[2], Ge Han [3], Xin Ma[1*], Huilin Chen[4,5]*, Truls Andersen[5], Huiqin Mao[6], Cuihong Chen[6], Haowei Zhang[1], Wei Gong[1,7]

[1] State Key Laboratory of Information Engineering in Surveying, Mapping and Remote Sensing, Wuhan University, Luoyu Road No.129, Wuhan 430079, China;

[2] School of Mathematics and Statistics, Wuhan University, Luoyu Road No.129, Wuhan 430079, China;

[3] School of Remote Sensing and Information Engineering, Wuhan University, Luoyu Road No.129, Wuhan 430079, China;

[4] Joint International Research Laboratory of Atmospheric and Earth System Sciences, School of Atmospheric Sciences, Nanjing University, Nanjing, China;

[5] Centre for Isotope Research, Energy and Sustainability Institute Groningen (ESRIG), University of Groningen, Groningen, Netherlands;

[6] Ministry of Ecology and Environment Center for Satellite Application on Ecology and Environment, Beijing, China;

[7] Electronic Information School, Wuhan University, Luoyu Road No.129, Wuhan 430079, China;

*Correspondence to*: Xin Ma (maxinwhu@whu.edu.cn); Huilin Chen (huilin.chen@rug.nl)

**Abstract.**

There are plenty of monitoring methods to quantify gases emission rate based on gases concentration samples around the strong sources. However, there is a lack of quantitative models to evaluate methane emission rate from coal mines with less priori information. In this study, we develop a genetic algorithm–interior point penalty function (GA-IPPF) model to calculate the emission rate of large point sources of $CH_4$ based on concentration sample. This model can provide optimized dispersion parameters and self-calibration, thus lowering the requirements for auxiliary data accuracy. During Carbon Dioxide and Methane Mission (CoMet) pre-campaign, we retrieve $CH_4$ emission rates from a ventilation shaft in Pniówek coal (Silesia coal mining region mine, Poland) based on the data collected by an UAV-based AirCore system and GA-IPPF model. And the concerned $CH_4$-emission rates are variable even in a single day, ranging from 621.3±19.8 to 1452.4±60.5 kg/hour on August 18, 2017 and from 348.4±12.1 to 1478.4±50.3 kg/hour on August 21, 2017. Results show that, $CH_4$ concentrations data reconstructed by the retrieved parameters are highly consistent to the measured ones. Meanwhile, we demonstrate the application of GA-IPPF in three gases control release experiments, and the accuracies of retrieved gases emission rates are better than 95.0 %. This study indicates that GA-IPPF model can quantify $CH_4$ emission rate from strong point sources with high accuracy.

## 1.Introduction

The release of $CH_4$ into the atmosphere during coal mining is very concerning because it contributes to increased atmospheric concentration of $CH_4$, one of the most important greenhouse gases and is a waste of resources (Cardoso-Saldana and Allen, 2020; Zhang et al., 2020). However, $CH_4$ emissions during coal mining are not always stable owing to different collection mode, manufacturing processes, weather fluctuations, as well as terrain effects (Nathan et al., 2015b). Bottom-up inventories can provide us with $CH_4$ emission rates from strong point sources or gridded $CH_4$ fluxes with different spatial resolutions, which play a great role in statistical analysis. However, the low temporal resolution of inventory data

does not allow us to obtain emission intensity from target sources instantaneously (Pan et al., 2021; Liu et al., 2020). With the development of different atmospheric $CH_4$ concentration measurement techniques, like Fourier spectrometer, differential absorption Lidar, AirCore system, and in-situ sensors, $CH_4$ emission rates from strong emission sources can be quickly quantified by top-down methods with high accuracy.

Greenhouse gases observing satellite (GOSAT) and TROPOspheric Monitoring Instrument (TROPOMI) are capable of obtaining the column concentration of $CH_4$ ($XCH_4$, ppb) with spatial resolution of 10 km×10 km and 5 km×7.5 km respectively. The regional $CH_4$ flux can be retrieved by assimilating the measured $XCH_4$ into an atmospheric dispersion model (Tu et al., 2022; Feng et al., 2016). Hyperspectral Precursor of the Application Mission (PRISMA) hyperspectral imaging satellite and GHGsat can detect increased $CH_4$ caused by strong emission sources with high spatial resolutions, and the comprehensive $CH_4$ emission can be quantified by integrated mass enhancement or cross-sectional flux method (Guanter et al., 2021; Varon et al., 2020). It plays a huge role in the analyzing methane emission rate from strong sources, but it has high requirements for satellites' detection track, that is, to monitor the methane distribution in the target area within coverage range (Schneising et al., 2020; Varon et al., 2019). Airborne sensors can fly at low altitudes to improve the acquisition of $CH_4$ concentration data and estimate $CH_4$ emission from strong sources by the cross-sectional flux method or the Gaussian dispersion method (Elder et al., 2020; Wolff et al., 2021a; Krautwurst et al., 2021). It enables repeated monitoring of emission sources in a large area in a short period of time, however, airborne experiments' cost is high and the flight tracks may be restricted by the aviation control policies. Ground-based eddy covariance sites can monitor agriculture and forestry ecology methane flux with high temporal resolution, such as mangrove ecosystem(Jha et al., 2014), larch forest in eastern Siberia(Nakai et al., 2020). Its accuracy is very high, but there is currently less monitoring of methane emissions from strong point sources using eddy covariance. When ground-based concentration sensors fixed in appropriate position, they have the advantage of continuously sampling gas concentration in downwind direction from the source. It will provide important dispersion data for methane emission quantification model at the enterprise level, but these sensors usually need to be carried on a vehicle platform to obtain methane concentration at different locations (Zhou et al., 2021; Robertson et al., 2017; Caulton et al., 2017). Ground-based differential absorption LIDAR can obtain the $CH_4$ profile concentration in different altitudes, whose data is suitable as the input of the emission-retrieval model (Shi et al., 2020a), but it has high requirements in terms of hardware performance and system stability (Shi et al., 2020b). An unmanned aerial vehicle (UAV) can reach any location rapidly around the $CH_4$ sources, which can sample $CH_4$ concentration with location information (Nathan et al., 2015b; Iwaszenko et al., 2021), when equipped with concentration sensors. It can acquire the distribution characteristics when sufficient concentration data are collected, which is beneficial to retrieving emission rate. The cost of UAV-based AirCore system is low and the process of its sample data is relatively simple, but the diffusion of methane emitted from strong sources may be sampled incompletely.

In 2017, we developed an UAV-based active AirCore system, which could sample spatial atmospheric $CO_2$, $CH_4$, and CO with high accuracy (Andersen et al., 2018), aiming to retrieve greenhouse gases emission from strong sources. The most urgent issue we need to address is developing an emission quantification model to make use of the advantage of AirCore, namely to collect data at different locations with a high degree of flexibility. This model should have less uncertainty in retrieved result and conform to the actual emission dispersion characteristics of the studied emission sources. Mass-balance method has been applied in determining $CH_4$ emissions based on UAV-based samples (Allen et al., 2018).

Emission rates calculated by this method contain large uncertainty because the main kernel is Kriging interpolation (Nathan et al., 2015a), which can cause obvious uncertainty in representing the actual feature of diffusion. The Gaussian dispersion model has also been applied in retrieving gas emission from strong sources (Shah et al., 2019; Ma and Zhang, 2016), and it is also used to model $CH_4$ diffusion in this study. However, existing emission-retrieval methods based on Gaussian dispersion model need priori information on key diffusion parameters (Nassar et al., 2021), which cannot be regarded as certain values in different circumstances. Moreover, the measurements accuracy of auxiliary meteorological data also has a great impact on $CH_4$ emission calculation.

To end this, we develop herein a model to overcome these shortcomings, named GA-IPPF, which combines the advantages of genetic algorithms (GA) and interior point penalty functions (IPPF). GA can model the fitness function as a process of biological evolution(Yuan and Qian, 2010), which can be used to calculate the potential solutions in Gaussian dispersion model. IPPF can find the minimum of the criteria in setting domain (Kuhlmann and Buskens, 2018), which can help us achieve global optimal solutions for concerned parameters. Finally, GA-IPPF can calculate the diffusion parameters without prior information and reduce the impact of meteorological data on the calculated $CH_4$-emission rate.

We introduce the structure of our developed GA-IPPF in detail in section 2. In section 3, we evaluate the performance of GA-IPPF in field campaign around a coal mine ventilation shaft by using AirCore system in 8 Flights. Then, we discuss the comparisons between different quantification methods for $CH_4$ emission, and evaluated the performance of GA-IPPF when the meteorological data are acquired from the fifth generation of ECMWF atmospheric reanalysis of the global climate (ERA5) database. In section 4, we validate the accuracy of GA-IPPF in Observing System Simulation Experiments (OSSE), and evaluate the uncertainty in retrieved emission rate of $CH_4$. Furthermore, we test the performance of GA-IPPF in quantifying emission rate based on three gases control release database.

## 2.Data and methods

### 2.1. Active AirCore System

The active AirCore system comprises a ~50 m coiled stainless-steel tube that works in conjunction with a micropump and a small pinhole orifice (45 $\mu m$) to sample air along the trajectory of a drone. If the pressure downstream of the orifice is more than half of that of the upstream (ambient) pressure, a critical flow through the orifice is obtained. This means that the flow rate depends only on two variables, namely, the air temperature and the upstream (ambient) pressure, both of which are monitored during the flight. After obtaining air samples during field campaigns, $CO_2$, $CH_4$ and CO collected by AirCore system would be analyzed by ground-based cavity ring down Spectrometer model G2401-m (Picarro). For $CH_4$, the accuracy of samples is ±0.02 parts per million (ppm). The active AirCore system is controlled using an Arduino-built data logger, which records the temperature inside the carbon fiber housing. It also records the ambient temperature, ambient pressure, relative humidity, and pressure downstream of the pinhole orifice to ensure that critical flow is achieved. The data logger also logs the GPS coordinates. The weight of the active AirCore system is ~1 kg. The active AirCore system is attached to a DJI Inspire Pro 1, which is capable of providing flights of ~12 min.

### 2.2. Meteorological measurements

A radiosonde (Sparv Embedded AB, Sweden, model S1H2-R) measures ambient temperature, ambient pressure, ambient relative humidity, wind speed, and wind direction. The detection range of the temperature sensor is –40 °C to +80 °C, with an accuracy of 0.3 °C. The pressure sensor has a detection range of 300–1100 mbar, with an accuracy of 1 mbar. The relative humidity sensor measures in the range of 0%–100%, with an accuracy of approximately 2%. Owing to the good connection between the

radiosonde and satellites, we assume that the uncertainty in the wind direction is low. The wind speed can be estimated within a range of 0–150 m/s, with an uncertainty of approximately 5%. If the wind-speed reading is less than 4 m/s, a minimum uncertainty of 0.2 m/s is given. The radiosonde is lifted by a ~30 L helium-filled balloon and is tethered onto a fishing line for easier retrieval after making a vertical profile.

## 2.3. Measurement Site

The Pniówek coal mine (49.975° N, 18.735° E) is a large mine in Pniówek, Silesian Voivodeship, Poland, which is 190 km southwest of the capital Warsaw, see Fig.1. It has a large coal reserve estimated to be about 101.3 million tons and coal production is about 5.16 million tons per year.

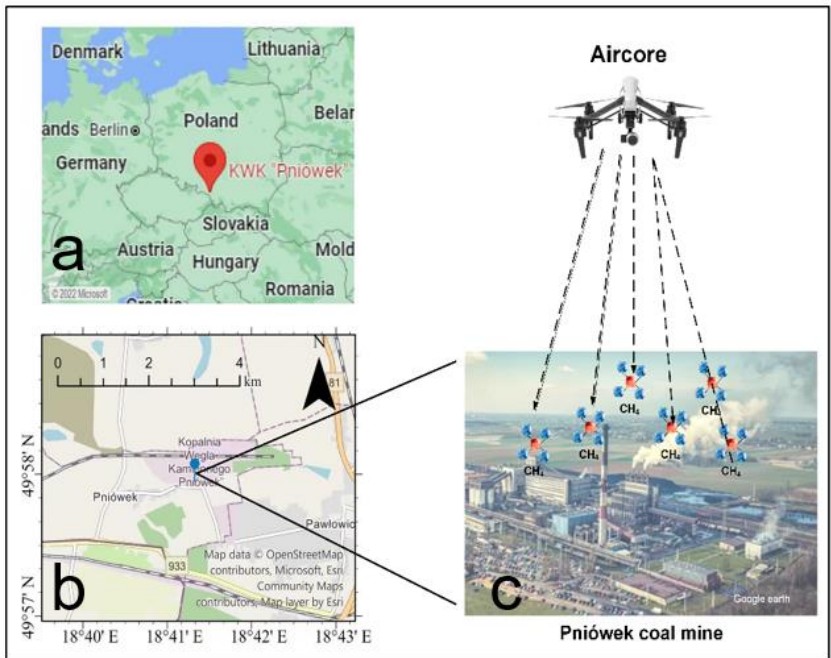

**Fig.1.** Pniówek coal mine; a. red mark represent the location of Pniówek coal mine in Poland; b. the surrounding circumstance of Pniówek coal mine, blue mark represent Pniówek coal mine; c. detailed layout of Pniówek coal mine, deep mine with shaft.

## 2.4 Emission retrieve model

### 2.4.1. Gaussian dispersion model

The Gaussian dispersion model was used to analyze the $CH_4$ fugitive from the coal mine in this work. The location of emission source is regarded as the coordinate origin; X-axis is the direction of the downwind, Y-axis is cross-wind direction, and Z-axis is the altitude above the ground. Based on the established coordinate system, the Gaussian plume can be modeled by Equation 1:

$$C(x, y, z) = \frac{q}{2\pi u \sigma_y \sigma_z} \exp(\frac{-(y)^2}{2\sigma_y^2}) \left\{ \exp(\frac{-(z-H)^2}{2\sigma_z^2}) + \alpha \cdot \exp(\frac{-(z+H)^2}{2\sigma_z^2}) \right\} + B \quad (1)$$

$$\sigma_y = a \cdot x^b \quad (2)$$

$$\sigma_z = c \cdot x^d \quad (3)$$

Where C is the concentration of $CH_4$ (g/m³), q (g/s) is the emission rate of $CH_4$ from coal mine, u is the mean wind speed around the stack (m/s), H is the effective stack height, $\sigma_y$ is the dispersion coefficient in the horizontal direction, $\sigma_z$ is the dispersion coefficient in the vertical direction, u is the wind speed (m/s), and B is the background concentration of $CH_4$. Moreover, $\alpha$ is the reflection index of the measurement phenomenon; and x, y, and z are the positions of the samples in the determined coordinate

system.

**2.4.2.GA-IPPF model.**

First, the genetic algorithm (GA) kernel calculates Q and other dispersion parameters with a first guess (Liu and Michalski, 2016). It guarantees that the unknown parameters are retrieved through the global optimum solution, as shown in Fig.2. Then, the results calculated by GA serve as initial input parameters and constraints in the IPPF model, and actual values of the concerned parameters are retrieved by IPPF. Detailed information can be found in S1 (supplement).

Based on the Gaussian dispersion model, auxiliary meteorological data, location information, and $CH_4$ samples, we determine the unknown parameters in equations 1 to 3 by using GA, including q, H, a, b, c, d, and α, in logical range constrained by lower boundary and upper boundary. First, the locations and concentration of $CH_4$ samples and wind serve as the initial input of equation 1. Then, the fitness value evaluates the applicability of the calculated parameters in each step. We define the fitness value as

$$F = \sum_{i=1}^{n} (C_m^i - C_s^i)^2 \tag{4}$$

$$C_s^i(x, y, z) = \sum_{i=1}^{n} \frac{q'}{2\pi u' \sigma_y' \sigma_z'} \exp(\frac{-(y)^2}{(\sigma_y')^2})\{\exp(\frac{-(z-H')^2}{(\sigma_z')^2}) + \alpha' \cdot \exp(\frac{-(z+H')^2}{(\sigma_z')^2})\} + B' \tag{5}$$

Where F is the fitness value; n is the total amount of concentration samples; $C_m^i$ is the sample $CH_4$ concentration; i is the number of samples; $C_s^i$ is the simulated concentration of $CH_4$ in the location of samples calculated by equation 5; and q′, u′, $\sigma_y'$, $\sigma_z'$, H′, α′, and B′ are the calculated $CH_4$-emission rate, wind speed, diffusion parameters, emission height, reflect index, and background $CH_4$ concentration, respectively, acquired from the "Mutation" in Fig.2. When f is less than the threshold value ($1\times10^{-5}$) of the fitness value, the corresponding parameters are treated as the results of output.

IPPF rebuilds the inequality constraint conditions to the unconstrained solution process. It forces the start point to satisfy the constraints, as shown in equation 6.

$$minF\left(x, r_k\right) = f(x) + r_k B(x) \tag{6}$$

Where f(x) is the unconstrained equation, and $r_k$ is the coefficient of the constrained equation B(x). When the solution parameters are out of the constraints, $r_k B(x)$ is large, thereby ensuring that the final solution is feasible under the inequality constraint conditions.

To obtain the inequality constraints, GA is repeated 10000 times, and the mean values of the calculated wind speed, wind direction, H, a, b, c, d, and α are treated as the initial input of IPPF model. The domains of H, a, b, c, d, and α are determined by two times the standard deviation of the corresponding results in GA. The constraint values of wind speed ($W_s$) and direction ($W_d$) are set according to the precision of actual measurements, m±σ,whereas m is the measured value of wind speed or wind direction, and σ is their measured precision. Actual B values are considered to be within 1800–2500 ppb. Then, the Pearson correlation coefficient (R) values of the actual samples and simulated values work as the criterion in the solution process of equation 7.

$$R = \frac{\sum_{i=1}^{n}\left(C_s^i - \overline{C_s}\right)\left(C_m^i - \overline{C_m}\right)}{\sqrt{\sum_{i=1}^{n}\left(C_s^i - \overline{C_s}\right)^2}\sqrt{\sum_{i=1}^{n}\left(C_m^i - \overline{C_m}\right)^2}} \tag{7}$$

The results are treated as the final retrieved values of the concerned parameters when the R reaches the maximum.

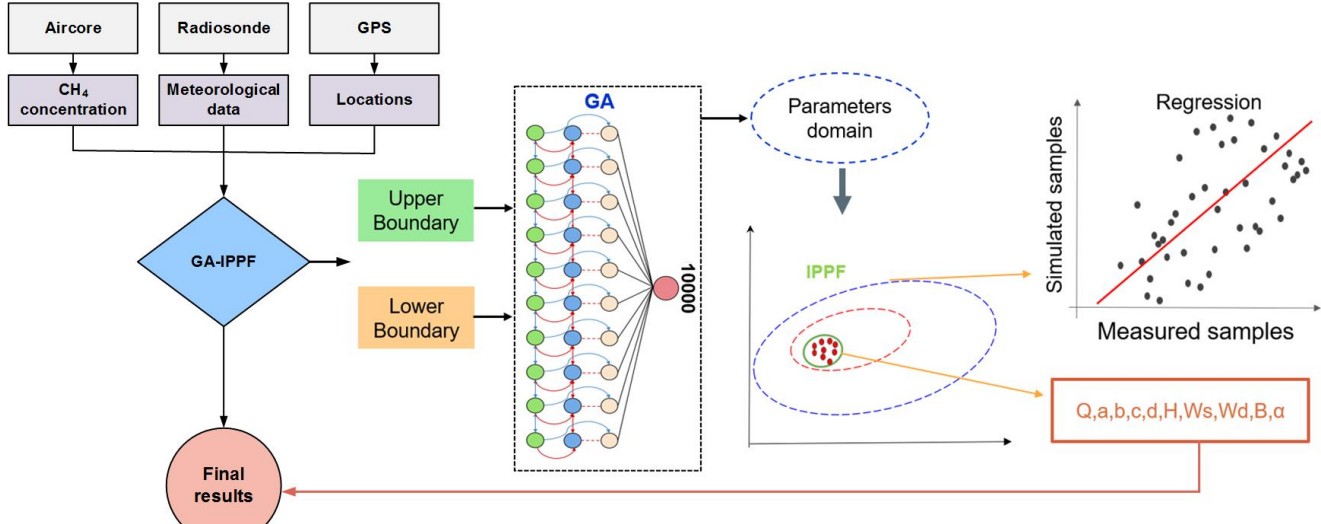

**Fig.2.** Flow chart of GA-IPPF model, including data inputs and process in each step.

**Uncertainty Analyses**

The GA-IPPF model will be calculated 1000 times repeatedly based on the collected samples of $CH_4$ concentration, then, the uncertainty and final retrieved emission rate could be defined by

$$\sigma = \sqrt{\frac{\sum_{i=1}^{N}(q_i - \bar{q})^2}{N}} \qquad (8)$$

$$q_r = \bar{q} = \frac{1}{N}\sum_{i=1}^{N}q_i \qquad (9)$$

$\sigma$ is the uncertainty of retrieved emission rate; $q_i$ is the i *th* retrieved emission rate, i=1,2,3…1000; $\bar{q}$ is the mean value of the $q_i$; N is 1000; $q_r$ is regarded as the value of retrieved emission rate. The values of other parameters (a,b,c,d,H,Ws,Wd,B,α) calculated by GA-IPPF are also defined in same principle.

**3. Results**

**3.1. Actual experiments**

As part of the Carbon Dioxide and Methane Mission (CoMet) pre-campaign,15 active AirCore Flights successfully collected data around a ventilation shaft of Pniówek coal mine on August 18, 2017 and August 21, 2017. The sample data in Flight 6 (18/8/2017) and Flight 15 (21/8/2017) were used to evaluate 210 the GA-IPPF model detailly, as shown in Fig. 3. Retrieved results of data collected by other Flights are presented in S2 (supplement).

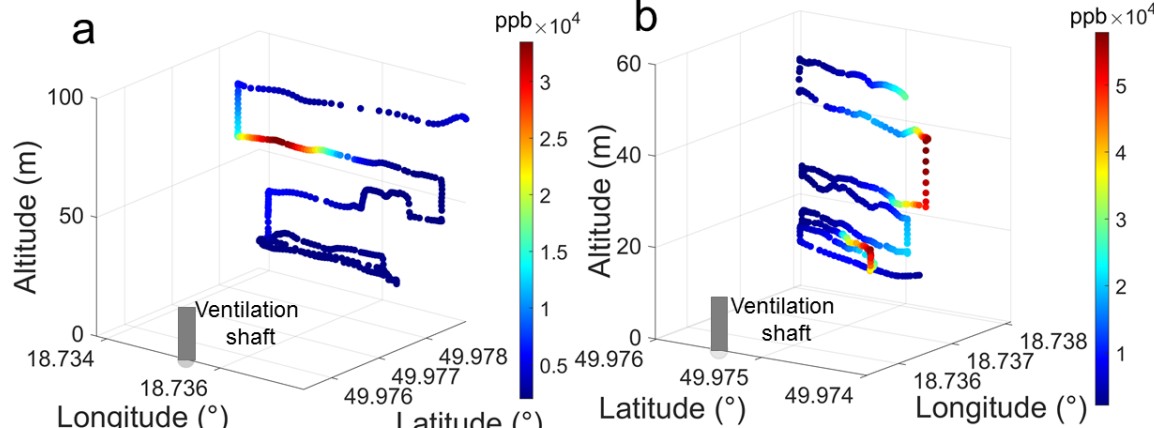

**Fig. 3.** Samples of CH$_4$ concentration in two Flights: (a) Flight 6 and (b) Flight 15.

In Flight 6, the AirCore system collected CH$_4$ from 0 m to 98 m around the ventilation shaft in a spiral
pattern, with a total of 376 samples, ranging from 1980.1 ppb to 49 113.9 ppb, and a measurement period
of 7 minutes. In Flight 15, the AirCore system collected CH$_4$ with a total of 400 samples, ranging from
2131.7 ppb to 57 265.3 ppb, and a measurement period of 9 minutes. Both Flights show high spatial
variability in CH$_4$ exhaust from ventilation shaft. Subsequently, we inputted the wind speed, wind
direction, location information, and CH$_4$ samples collected from Flights into the GA-IPPF model. To
express the final retrieved emission (Q) in g/s, the dry-air mixing ratio of CH$_4$ (ppb) is transformed into
mass concentration m (mg/m$^3$) as follows:

$$m = C \cdot \frac{M_{CH4}}{M_{Air}} \cdot 10^{-3} \tag{10}$$

Where M$_{CH4}$ is the molar mass of CH$_4$, and M$_{air}$ is the molar mass of air.

The retrieved results are shown in Table 1, the uncertainty is presented in Discussion in detail. Notably,
the emission height in Flight 15 was larger than that of Flight 6, which might be caused by the difference
in thermal energy and vertical wind speed of the two flights. The background concentrations of CH$_4$ were
1.43 and 1.41 mg/m$^3$ in Flights 6 and 15, respectively, which show little difference. The dates of the two
Flights were very close, so the background concentration of CH$_4$ in two days had nearly the same seasonal
characteristics. The exhaust gases of coal mine were emitted through the ventilation shaft with effective
emission heights of 58.4 and 35.5 m, respectively.

To evaluate the rationality of the retrieved results, these parameters are used to simulate CH$_4$ diffusion
from the ventilation shaft according to equation 1. The comparison between simulated CH$_4$ concentration
data and actual samples in the same locations is shown in Fig.4.

Table 1. Results calculated by GA-IPPF model

| Parameters | Flight 6 | Flight 15 |
|---|---|---|
| Initial wind speed (m/s) | 2.8 | 3.2 |
| Initial wind direction (°) | 310 | 125.4 |
| Emission intensity (kg/hour) | 693.7±20.2 | 958.9±42.4 |
| Wind speed (m/s) | 2.83±0.2 | 2.4±0.3 |
| Wind direction (°) | 349.6°±1.2 | 128.1±0.8 |
| a | 0.60±0.01 | 0.31±0.01 |
| b | 0.73±0.02 | 0.95±0.01 |
| c | 0.2±0.01 | 0.08±0.01 |

| | | |
|---|---|---|
| d | 0.68±0.01 | 0.94±0.02 |
| B (mg/m³) | 1.43±0.01 | 1.41±0.01 |
| Emission height (m) | 58.4±2.3 | 35.5±1.8 |
| Reflection index | 0.90±0.01 | 1.0±0.01 |

Then, we also calculate the difference between the actual measured samples and simulated ones as

$$D_c = C_s - C_m \qquad (11)$$

Where $D_c$ is the difference of $CH_4$ concentration data between actual measured and simulated ones, $C_s$ is simulated $CH_4$ concentration (mg/m³), and $C_m$ is measured $CH_4$ concentration (mg/m³).

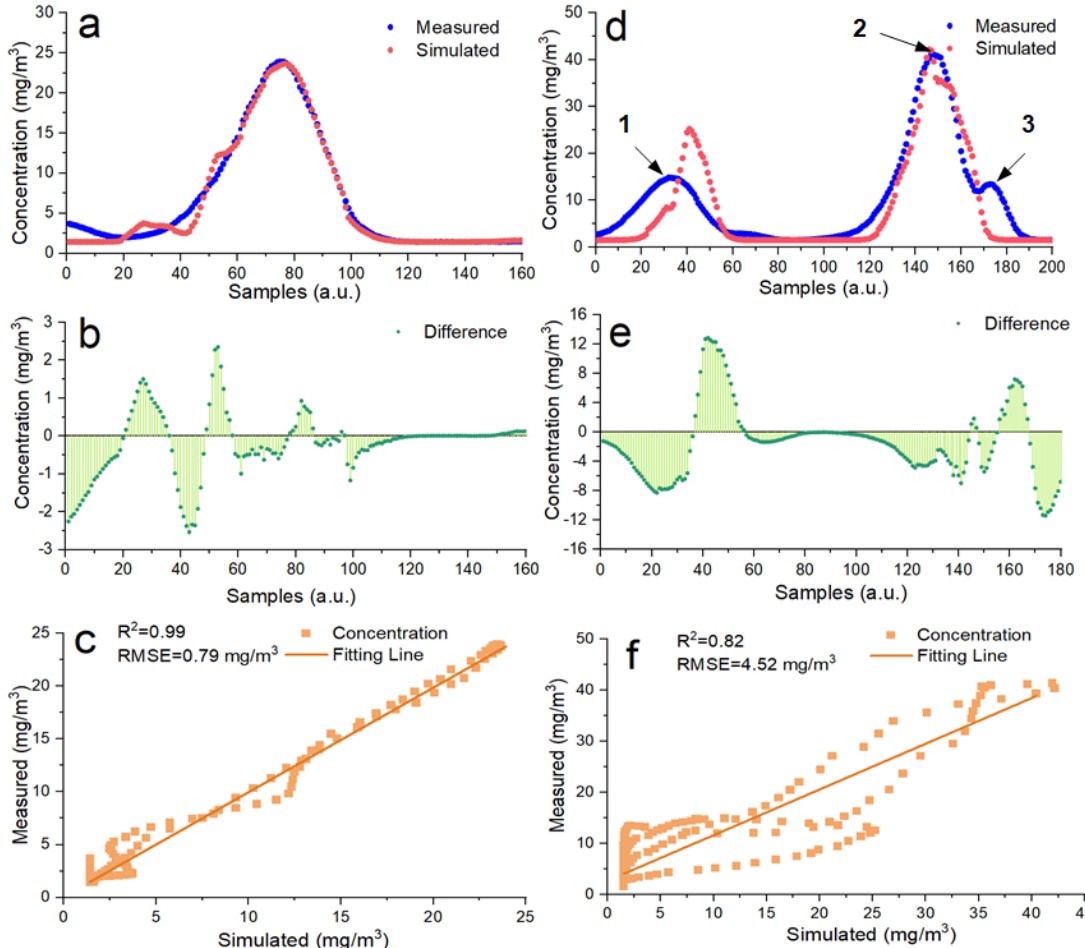

**Fig. 4.** Comparison between the measured samples and the simulated ones based on the parameters in Table 1: (a). Flight 6 and (d) Flight 15. The difference of simulated $CH_4$ concentration data and actual measured ones: (b) Flight 6 and (e) Flight 15. Correlation Analysis: (c) Flight 6 and (f) Flight 15.

Fig 4 shows that the simulated $CH_4$ concentration is high consist with actual sampled ones in two Flights. In Flight 6, the largest value of sampled $CH_4$ concentration is 23.92 mg/m³, while the corresponding simulated one is 22.45 mg/m³, relative error is only 0.2 %. It is worth noting that it exists three peaks in Flight 15, mainly occur at the altitudes of about 16 m, 25 m and 40 m, see S3 in supplement. Fig.4 (d) shows the simulated $CH_4$ concentration data around the 1th and 3th peak are not better than that around the 2th peak. Because GA-IPPF method can assign more weights to the samples with higher concentration (NO.120 to 180 in Flight 15) to get the global optimal solution of the unknown parameters, which leads to lower fitness to simulated $CH_4$ concentration around the 1th and 3th peaks. Values of Dc are ranging

from -2.50 to 2.35 mg/m$^3$ in Flight 6, which are lower than that in Flight 15. $R^2$ between simulated $CH_4$ concentration and actual sampled ones are larger than 0.8 in two Flights, root mean square errors (RMSEs) are 0.79 mg/m$^3$ and 4.52 mg/m$^3$ respectively. The simulated $CH_4$ concentration in the other Flights are seen in S2. In summary, the tendency of the simulated $CH_4$ concentration data remains consistent with that of the actual samples in Flight.

### 3.2 Comparison with other methods

To investigate the difference between our proposed emission model and the others, three methods were applied to estimate $CH_4$ emission in all Flights, including mass-balance approach, nonlinear least square fit (NLSF), and facility emission.

Mass-balance approach quantifies $CH_4$ emission by calculating the cross-sectional flux perpendicular to the wind direction (Krings et al., 2018). First, a two-dimensional plane is selected according to the amount of $CH_4$ samples. Second, the two-dimensional plane is divided into a grid of equal spatial resolution. Third, $CH_4$ samples are regarded as original points to interpolate full grids defined by the Kriging interpolation scheme (Mays et al., 2009). Finally, the emission rate of the $CH_4$ source is calculated by

$$F_{(CH4)} = \iint v \sin(\alpha) \cdot (C_{(x,z)} - C_{bg}) dxdz \qquad (12)$$

Where v is the wind speed, $\alpha$ is the angle between wind direction and the two-dimensional plane, $C_{(x,z)}$ is the density of $CH_4$ in each grid, and $C_{bg}$ is the background of $CH_4$ in each grid. The uncertainty analyses of this method are detailed in Nathan et al. (Nathan et al., 2015a).

NLSF and the combination of NLSF with Gaussian diffusion model are also extensively used for point-source emission retrieval (Zheng et al., 2020; Wolff et al., 2021b). In this study, NLSF is used to estimate Q in each Flight by fitting the unknown parameters in equation 1.

Andersen et al. also developed an inverse Gaussian approach to quantify $CH_4$ emissions from coal mine ventilation shaft based on the same Flights (Andersen et al., 2021). Firstly, the Gaussian dispersion is built as

$$C(x, y, z) = \frac{q}{2\pi u \sigma_y \sigma_z \cos(\theta)} \exp(\frac{-(y)^2}{2\sigma_y^2}) \left\{ \exp(\frac{-(z-H)^2}{2\sigma_z^2}) + \exp(\frac{-(z+H)^2}{2\sigma_z^2}) \right\} \qquad (13)$$

Where $\theta$ is the angle between the wind direction and the perpendicular line of the flight trajectory. This model does not include the item of background of $CH_4$. Furthermore, $\sigma_y$ and $\sigma_z$ are treated as certain values in equation 11.

Facility-emission data and hourly $CH_4$ emission from shaft are calculated by measuring raw $CH_4$ concentration and air flux through the shafts, following the equation below

$$Q_{Inventory} = \frac{P \cdot V_{flow}}{R \cdot T} \rho \qquad (14)$$

Where $V_{flow}$ is the volumetric flow rate of $CH_4$ in m$^3$ s$^{-1}$, given by the air flow rate (scaled by a constant factor of 0.95 to account for the ~5% additional air flow not coming from the ventilation shaft) multiplied by the $CH_4$ concentration, and P, R, T, $\rho$ are the atmospheric pressure in Pa, the universal gas constant in J mol$^{-1}$ K$^{-1}$, the ambient temperature in K, and the molar density of $CH_4$ in g mol$^{-1}$ (16.043 g mol$^{-1}$), respectively.

$CH_4$ emission rates from ventilation shaft estimated by hourly facility-emission data for 18 August 2017 and 21 August 2017 are 1655.3 ±479.45 and 913.2 ±285.4 kg/hour, respectively, as shown in Fig. 5.

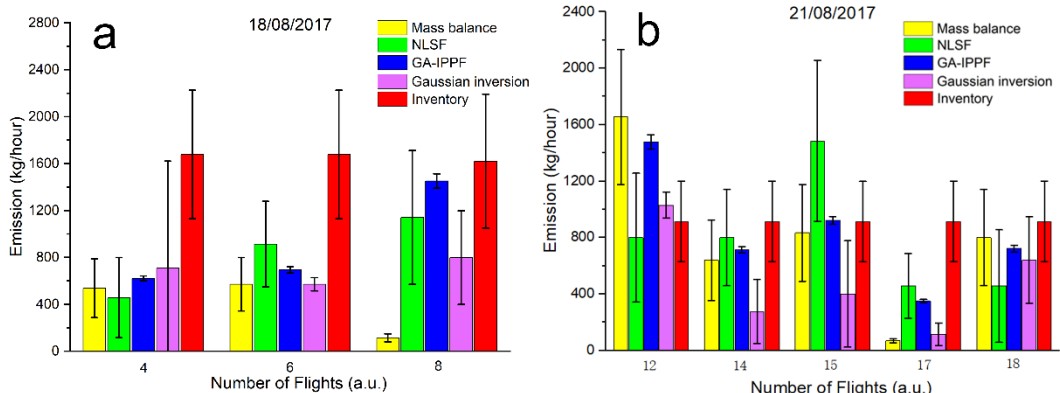

**Fig. 5.** Quantified CH$_4$ emission by different methods based on the collected data: (a) August 18, 2017 and (b) August 21, 2017. CH$_4$ emission rates from ventilation shaft calculated by Mass balance and Inverse Gaussian refer to Andersen et al.

As shown in Fig. 5, Flights 4, 6, and 8 were measured on 18 August 2017, whereas Flights 12, 14, 15, 17, and 18 were measured on 21 August 2017. Fig.5 (a) shows that the CH$_4$-emission rates calculated by mass balance are smaller than the inventory estimation in all Flights. In Flight 8, q retrieved by mass balance is extremely lower than those quantified by other methods, whereas q retrieved by GA-IPPF model (1478.4±50.3 kg/hour) shows only a slight difference from the inventory. As shown in Fig.5 (b), CH$_4$ emissions retrieved by mass balance, inverse Gaussian, and GA-IPPF model are overestimated compared with the inventory in Flight 12. Mass balance and inverse Gaussian method also show obviously underestimated q in Flight 17. Estimations of retrieved CH$_4$ emission in Flight 18 show consistency among methods of mass balance, GA-IPPF, and inverse Gaussian. The CH$_4$ emission rate of coal generally has significant variability in each measurement, even on the same day. Mass balance is very sensitive to the size settings of grids, and both height and length settings can affect the concentration distribution across the cross-section. NLSF has a high-accuracy requirement for wind measurements, and errors on these measurements have a linear influence on the final emission estimation. Notably, the standard errors of q quantified by GA-IPPF are always the least among these methods, indicating the stability of the model we developed. And we also simulated 2-D CH$_4$ plume from the ventilation shaft in Flight 6 and Flight 15 based on different methods, seen S4.

**3.3 Application of Reanalysis meteorological database in GA-IPPF model**

Wind speed and wind direction acquired by the radiosonde or weather station are two main parameters in GA-IPPF. However, additional sensors are bound to increase the cost and difficulty during actual CH$_4$-emission measurements. To explore the possibility of weather reanalysis data instead of actual wind measurement by sensors, we use 10 m U and V wind components from the ERA5 meteorological reanalysis database (spatial resolution is 0.1°×0.1°, and temporal resolution is 1 h) developed by the European Centre for Medium-range Weather Forecast (Hersbach et al., 2020) to evaluate GA-IPPF model. However, the wind directions from ERA obviously differed from the actual measurements during the Flights. Hence, we determine the wind direction by using the CH$_4$ samples, for example, the line between the shaft and the location of the maximum value of samples in the same heights was treated as the downwind direction, whose uncertainty was set as 50°. Wind speed from ERA is used for the CH$_4$-emission calculation, and the uncertainty was supposed as 2 m/s. Even if the initial wind speed and wind direction obviously differed between the two sources, the GA-IPPF model adjusted them into reasonable ranges. The results of q retrieved by two meteorological data sources during all Flights were evaluated, as shown in Table 2.

Table 2. CH$_4$ emission retrieved by two meteorological data sources

| Flights | Measured (kg/hour) | ERA5 (kg/hour) |
|---|---|---|
| 4 | 621.3±19.8 | 672.8±25.2 |
| 6 | 693.7±26.2 | 726.6±37.3 |
| 8 | 1452.4±60.5 | 1597.4±82.7 |
| 12 | 1478.4±50.3 | 1526.8±64.9 |
| 14 | 712.6±21.2 | 597.8±32.7 |
| 15 | 922.9±27.4 | 874.7±37.4 |
| 17 | 348.4±12.1 | 390.1±14.2 |
| 18 | 722.0±24.8 | 784.1±27.4 |

Table 2 shows that the values of quantified q between the two meteorological sources are within 20% in the same Flight. The standard errors of q retrieved by the ERA5 database are larger than those from actual measurements, wind data acquired from ERA5 database perhaps be treated as alternative input parameters in GA-IPPF model if no meteorological instruments are equipped in field experiments.

We also explore the reason that little difference of the calculated emission rates by the two different sources of meteorological data. The concerned parameters in Flight 6 and Flight 15 calculated based on ERA5 meteorological data are presented in Table 3.

Table 3. Parameters retrieved by GA-IPPF through ERA5 database

| Parameters | Flight 6 | Flight 15 |
|---|---|---|
| Initial wind speed (m/s) | 2.5 | 2.4 |
| Initial wind direction (°) | 300 | 120 |
| Emission intensity (kt/hour) | 726.6±37.3 | 898.7±52.1 |
| Wind speed (m/s) | 2.5±0.4 | 2.2±0.3 |
| Wind direction (°) | 349.4°±2.1 | 128.1±0.4 |
| a | 0.60±0.02 | 0.30±0.01 |
| b | 0.73±0.03 | 0.97±0.02 |
| c | 0.40±0.02 | 0.07±0.02 |
| d | 0.57±0.02 | 0.96±0.01 |
| B (mg/m$^3$) | 1.43±0.01 | 1.41±0.01 |
| Emission height (m) | 59.2±3.1 | 35.1±2.7 |
| Reflection index | 0.94±0.02 | 0.90±0.03 |

The initial wind speed and wind direction in Table 3 are obviously different from those in Table 1.

However, the retrieved wind directions are nearly the same based on the two sources of meteorological data. Retrieved diffusion parameters and emission heights are also show less difference in two Tables (Table 1 and Table 3). It is worth noting that the wind speed and reflection index can be adjusted to reach the global solution by GA-IPPF model, which leads to little bias in retrieving emission rate.

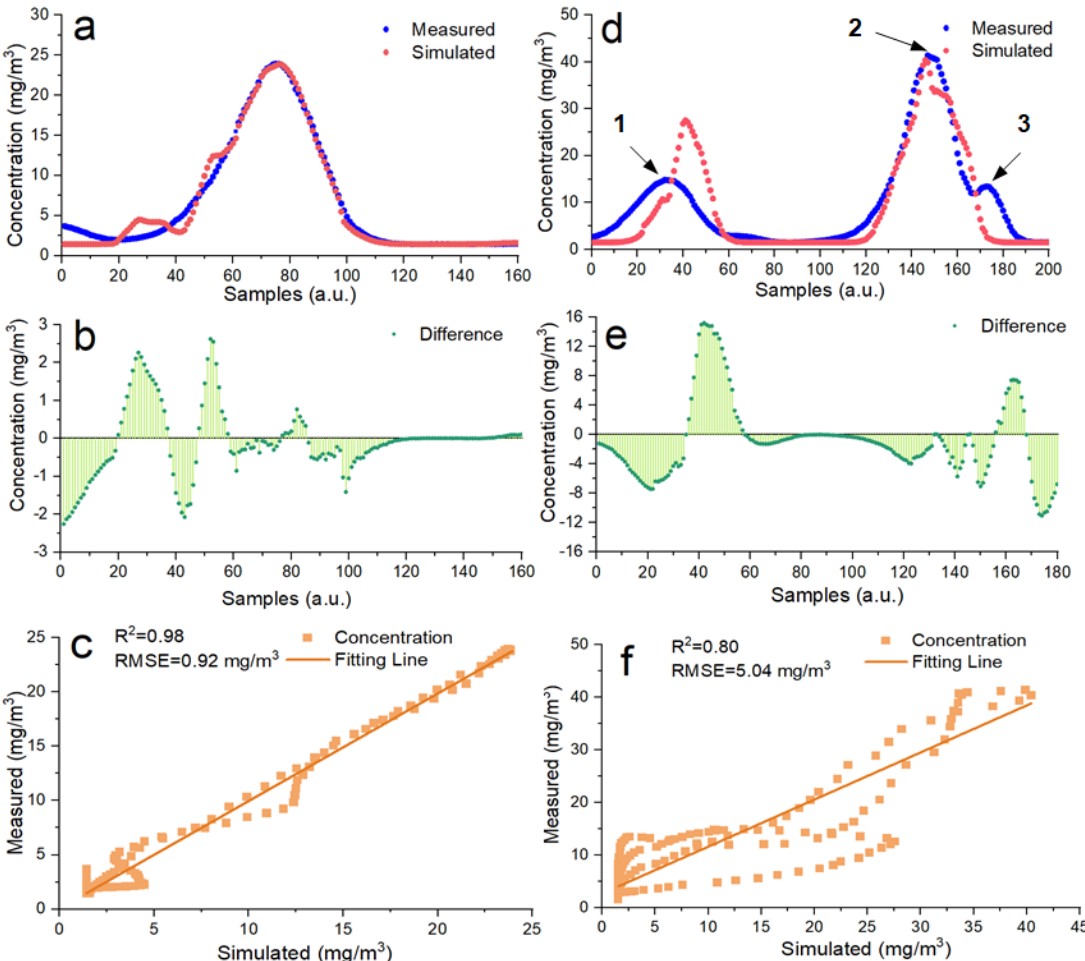

**Fig. 6.** Comparison between the measured samples and the simulated ones based on the ERA5 meteorological data: (a). Flight 6 and (d) Flight 15. The difference of simulated $CH_4$ concentration data and actual measured ones: (b) Flight 6 and (e) Flight 15. Correlation Analysis: (c) Flight 6 and (f) Flight 15.

The tendency of simulated $CH_4$ concentration data in the two Flights are similar to that in Fig.4. What's more, both $R^2$ and RMSE between simulated $CH_4$ concentration data and actual measured ones in both Flights show less difference with that in Fig.4. Values of $D_c$ shown in Fig.6 (b) are ranging from -2.25 to 2.62 mg/m³, which is nearly the same as the result in Fig.4 (b). Values of $D_c$ shown in Fig.6 (e) are ranging from -11.04 to 15.21 mg/m³, while -11.3 to 12.85 mg/m³ in Fig.4 (e). Because the difference between actual measured wind speed and ERA 5 speed is 0.8 m/s in Flight 15, which is larger than that in Flight 6 (0.3 m/s). In summary, GA-IPPF can still simulated reasonable diffusion of $CH_4$ through ERA5 wind data.

**4.Discussion**

**4.1. Validation of performance of GA-IPPF model through OSSEs**

Firstly, the dispersion of $CH_4$ emission from a strong point source was simulated by equation 1 using the dispersion parameters shown in Table 4. To make the simulations close to the actual measurement scenarios, random errors were added to the $CH_4$ concentration samples (0.5 %), wind speed (± 0.3 m/s), and wind direction (± 20°). Then, the simulated flight track of UAV was conducted in crossing section (300 m to strong source), in Fig 7. The spatial resolution of the supposed samples is set as 10 m, and 99 samples are selected from the simulated dispersion to represent the data acquired by the UAV-based

AirCore. Then, the concerned parameters are retrieved by the GA-IPPF method based on the above
assumptions. Simulations are repeated 10 000 times, and the average values of the corresponding
parameters were treated as the "Retrieved" results in Table 4.

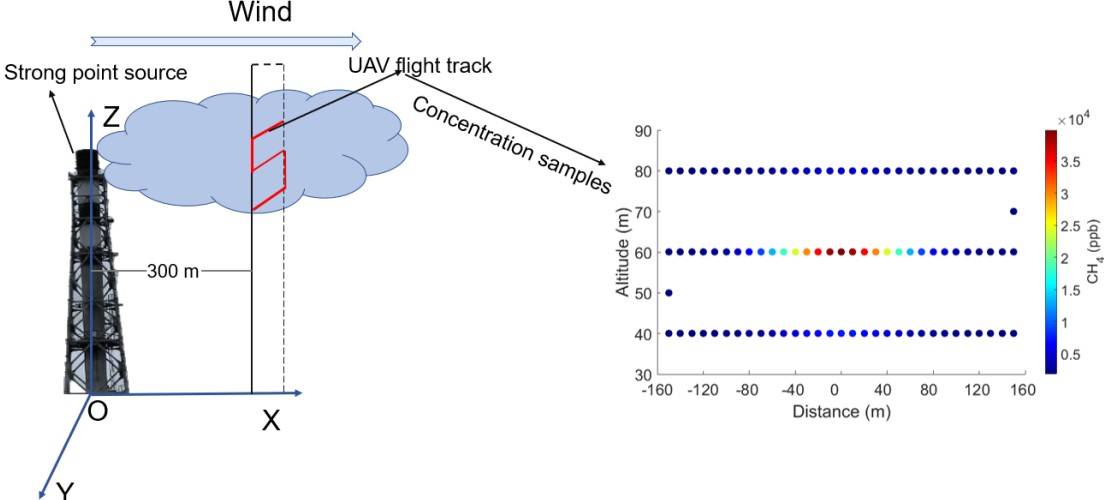

**Fig. 7.** Rectangle represents crossing section perpendicular to wind direction, covering a distance of
300 m near the point source; Red line represents simulated flight track of UAV-based AirCore system;
colored points represent the CH4 concentration samples in OSSEs, totally 99.

Table 4. The parameters setting in dispersion simulation and the retrieved results by GA-IPPF

| Parameters | Lower boundary | Upper boundary | Actual | Retrieved |
|---|---|---|---|---|
| Emission intensity (g/s) | 0 | 100000 | 180 | 180.2±0.02 |
| Wind speed (m/s) | 0 | 100000 | 3 | 3±0.01 |
| Wind direction (°) | 70 | 110 | 90 | 90±0.10 |
| a | 0 | 1000 | 0.6 | 0.6±0.02 |
| B | 0 | 1000 | 0.7 | 0.7±0.02 |
| c | 0 | 1000 | 0.2 | 0.2±0.01 |
| d | 0 | 1000 | 0.6 | 0.6±0.01 |
| B (ppb) | 1700 | 2500 | 1900 | 1900±2.7 |
| Emission height (m) | 0 | 150 | 50 | 49.8±1.1 |
| α | 0 | 1 | 0.9 | 0.91±0.01 |

"Actual" means the set values of parameters, and "Retrieved" means the average values of parameters
retrieved by GA-IPPF model through 10 000 times of simulation.

As shown in Table 1, q retrieved by GA-IPPF has only 0.11% bias compared with the set values.
Emission height only has 0.2 m bias in terms of the set one, and the uncertainty is only 0.4% to 50 m.
Other retrieved parameters also show high consistency with the original settings.

**4.2. Stability analyses**

The necessary input parameters in GA-IPPF contain meteorological data (wind speed and wind direction),
accuracy of CH4 samples, and amount of CH4 samples. In equation 1, wind speed has a nearly linear
relationship with the emission estimation. Wind speed is also an important factor that determines
atmospheric stability according to the Pasquill–Gifford method (Venkatram, 1996) as it affects the
diffusion parameters of $\sigma_y$ and $\sigma_z$. The coordinate is built according to the wind direction, which is defined
as the plane coordinates of CH4 samples. According to equations 2 to 3, errors in wind-direction

measurement lead to wrong $\sigma_y$ and $\sigma_z$ on each position of samples. $CH_4$ samples are the most important factors to determine the Gaussian diffusion. The accuracy of samples influences the judgment of "fitness" in the GA process. More samples collected in different positions help rebuild the spatial-distribution characteristics of the plume, because this provides larger possibility for fitting process in IPPF and helps determine the optimum solution. To evaluate the influence of errors in the measurements of necessary
parameters on the final retrieved results, the same settings in Table 4 are used as actual results. The performance of the GA-IPPF model with additional random errors in each parameter was simulated 10 000 times, as shown in Fig. 8.

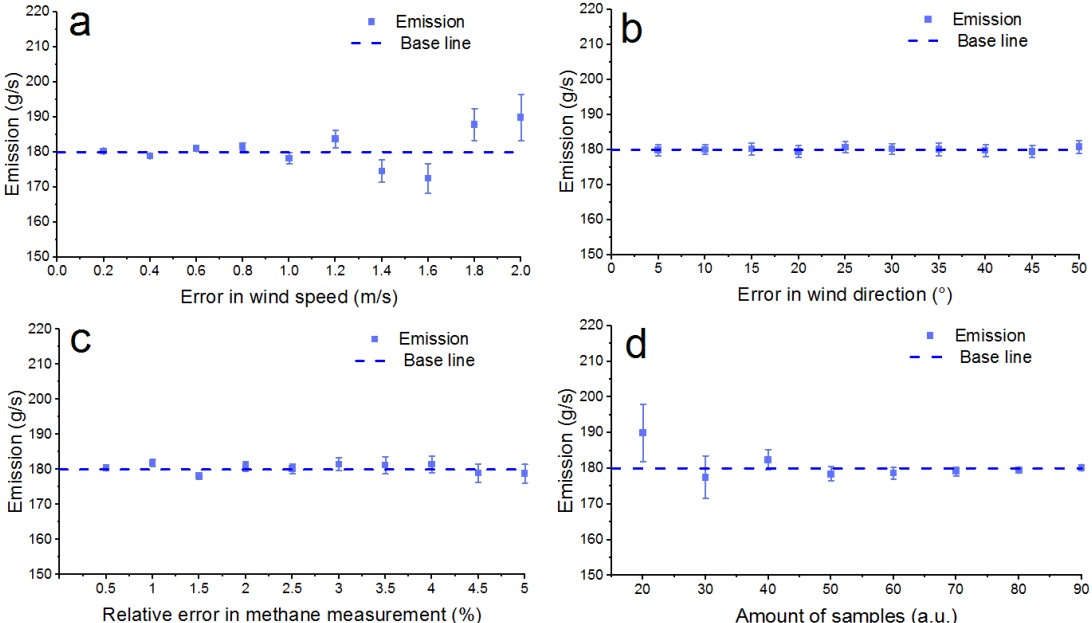

**Fig. 8.** Influence of accuracy of parameters on retrieved emission results. The baseline represents the
390 emission rate setting of $CH_4$, 300 g/s: (a) wind speed, with additional error ranging within 0.2–2 m/s and an interval of 0.1 m/s, (b) wind direction, with additional error ranging within 5°–50° and an interval of 5 °, (c) accuracy of $CH_4$ samples, with additional error ranging within 0.5%–5.0% and an interval of 0.5%, and (d) amount of $CH_4$ samples, randomly selected as 20–90 among the defined 99 samples.

In Fig.8 (a), the mean value of q retrieved by GA-IPPF is nearly the same as the baseline if the error
in wind speed is less than 0.4 m/s. It occurs to maximum retrieved emission bias (10.2 g/s) to the baseline when 2 m/s error in wind speed. Fluctuation of q occurs obviously if the error in wind speed exceeds 0.4 m/s. The standard errors of q are positively correlated with the values of errors in wind speed, indicating that the accuracy of wind-speed measurements largely influence the stability of the GA-IPPF model. This model has a self-adjustment function for wind speed; for example, when the initial wind speed is 3 m/s,
the maximum standard error of q is only 6.6 g/s (3.7% to the 180.0 g/s) when the additional error of wind is 2.0 m/s (66.7% to 3.0 m/s).

The retrieved q shows less sensitivity to errors in wind direction (see Fig.8 (b)). When errors in wind direction are 5° to 40°, all biases of q are within 0.7 g/s and the standard errors are around 1.6 g/s. Wind direction determines the spatial location of the sampling point, and wrong location information leads to
405 distinct errors in emission estimation. GA-IPPF shows highly accurate ability to obtain the global optimum solution in wind direction.

Sampling accuracy has small impact on the retrieved q within different settings in $CH_4$ samples' accuracy, see Fig.8 (c). Standard deviation is positively correlated with errors in $CH_4$ measurements. The

standard deviation is 2.6 g/s when the measurement error reaches 5.0 %. Notably, the uncertainty of $CH_4$ samples measured by UAV-based AirCore system is far less than 5.0 %. The UAV-based AirCore system can acquire more than 99 $CH_4$ samples in actual feasible measurements, therefore, it is believed that accuracy of $CH_4$ samples (>95.0 %) collected by the AirCore system bring less influence in theory.

The number of measurement points obviously influences the final accuracy of q by the GA-IPPF model (see Fig.8 (d)). It has a bias of 9.7 g/s to 180.0 g/s when n is 20. The accuracy of q and the standard error are negatively correlated with n which provides the number of criterion for the fitting process in the retrieval model. Hence, n directly influences the retrieved results. The AirCore system has the advantage of continuous sampling during flight, which integrates the atmospheric signals along the flight path and helps reduce the uncertainty in the retrieved q. Besides, the smoothing of the atmospheric signal also reduces the spatial resolution of the measurements, which needs to be considered during the optimization.

IPPF can suitably solve the problem of inequality constraints, and the calculated solution guarantees the calculated parameters to be within the feasible region. In this section, the performance of the GA-IPPF model and the influence of the four key input parameters are discussed.

**Suggestions for quantifying emission rate through UAV-based AirCore system**

1. Meteorological instruments should be equipped when collecting concentration samples to acquire wind speed, wind direction, humidity and atmosphere pressure.

2. The wind speed should be greater than 2.0 m/s.

3. During actual experiments, after the stable wind speed and wind direction are measured, the UAV-based AirCore system will start its concentration collection, and the system should try to fly along the cross section perpendicular to the wind direction.

These criterions are also determined the analyzed 8 Flights from the total 15 Flights.

**Application of GA-IPPF**

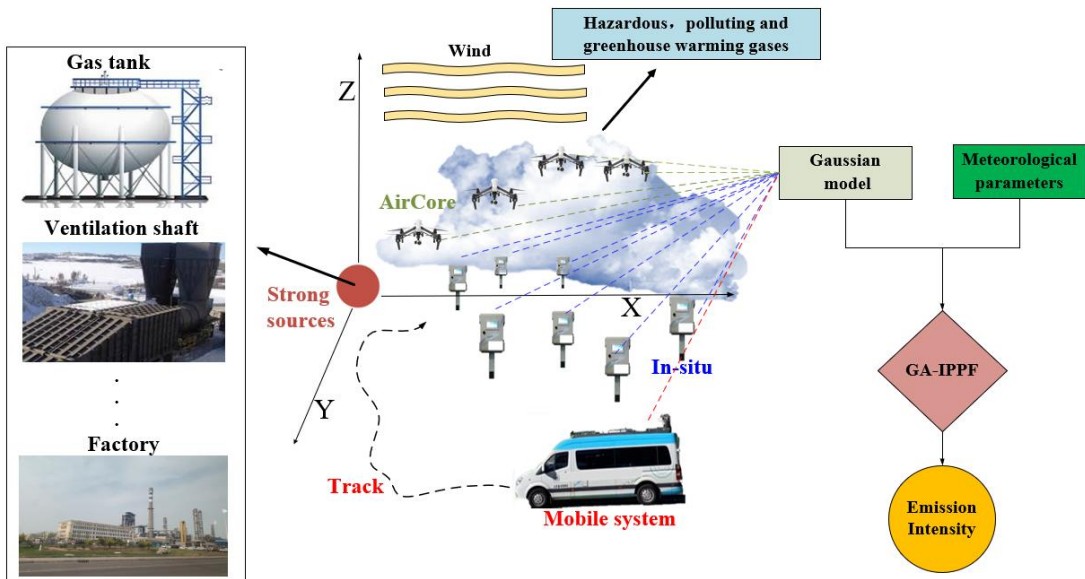

**Fig.9.** Application of GA-IPPF in quantifying emission source of gases through different sample systems, including UAV-based AirCore system, ground-based In-situ network and mobile collection system.

GA-IPPF, works as an emission gas quantification method, which can achieve rapid real-time monitoring of methane leakage caused by landfills, chemical plants and other strong sources. In theory, the recommended model is applicable not only to UAV-based AirCore system, but also to other sample systems which can measure gases concentration and location information. Each country's environmental

monitoring department may have built gases sample equipment based on different platforms, including
UAV, vehicles, and ground-based in-situ stations. These systems may not only monitor greenhouse gases
like $CO_2$ and $CH_4$, as well as polluting and harmful gases. Therefore, we demonstrate the application of
GA-IPPF in quantifying gases emission based on different gases concentration collected systems in
actual experiments.

**Emission Estimates in control release experiment**

To evaluate the performance of GA-IPPF in control release experiments, we quantify the gases
emission rates in release experiment through different gases sample systems, including UAV-based
AirCore system, mobile sampling system and ground-based in-situ network. Detailed introduction of the
concerned release experiment are as follows:

**Agrar Hauser control release**

This $CH_4$ release experiment was conducted on Agrar Hauser field near Dübendorf,
Switzerland(Morales et al., 2022). The controlled $CH_4$ was release from an artificial source, 50 L high-
pressure cylinder with a height of 1.5 m. Meteorological information were acquired by 3D anemometers
around the emission source. UAV-based sample systems used in these release experiments contained two
sensors, Quantum cascade laser spectrometer (QCLAS) and active AirCore. It carried series active
measurements from 23 February to 14 March 2020.There was no other $CH_4$ source around Agrar Hauser
field and the topography was flat. In this section, active AirCore $CH_4$ samples on 12 march 2020 (312_01)
were chosen to use GA-IPPF to quantify methane release rate.

**EPA methane control release**

Environmental Protection Agency (EPA),USA developed OTM 33A method to quantify oil and gas
leakage based on mobile measurement platforms(Brantley et al., 2014), which consisted $CH_4$ in-situ
sensor (G1301-fc cavity ring-down spectrometer (Picarro)), a collocated compact weather station and a
Hemisphere Crescent R100 Series GPS system. The accuracy of in-situ sample was within $\pm5\%$, and
in-situ sensor was implemented at height of 2.7 m based on vehicle. Weather station provided
atmospheric temperature, pressure and humidity, as well as 3-D wind direction and wind speed. A 99.9%
$CH_4$ high pressure cylinders was used as the gas supply to simulate the $CH_4$ leakage source. EPA
published total 20 experiments of control releases to evaluate OTM 33A method.

**Prairie Grass emission experiment**

Prairie Grass emission experiment was mainly conducted to evaluate the diffusion of $SO_2$ from point
source under different meteorological circumstances (Barad et al, 1958). The height of emission source
was 0.46 m, and all in-situ sensors were set at heights of 1.5 m. $SO_2$ concentration was sampled by the
in-situ network at the radius of 50 m, 100 m, 200 m, 400 m and 800 m around the source. Samples in
R57 release (10-minute sampling periods), totally 94, were selected to quantified $SO_2$ emission rate from
release instrument. The reported emission rate of $SO_2$ in R57 was 105.1 g/s, and the samples collected at
the radius of 800 m were neglected in this discussion because of their very small quantity. The reported
wind speed was 4.85±1 m/s, wind direction was 184±10°.

**Table 4 Performance of GA-IPPF model in different control release experiments**

| Database | Number | Gas | Release rates (g/s) | Retrieved by GA-IPPF (g/s) |
|---|---|---|---|---|
| Agrar Hauser | 312_01 | $CH_4$ | 0.31±0.03 | 0.3±0.03 |
| EPA | STR_6061411_01 | $CH_4$ | 0.60 | 0.57±0.04 |
| Prairie Grass | 57 | $SO_2$ | 101.5 | 104.7±3.7 |

Table 4 shows the emission rates and uncertainties through GA-IPPF in control release experiments,
and the reported emission rates. The average difference between retrieved emission rates and reported

ones is 3.8 %, which indicates the high accuracy of GA-IPPF in quantification estimation.

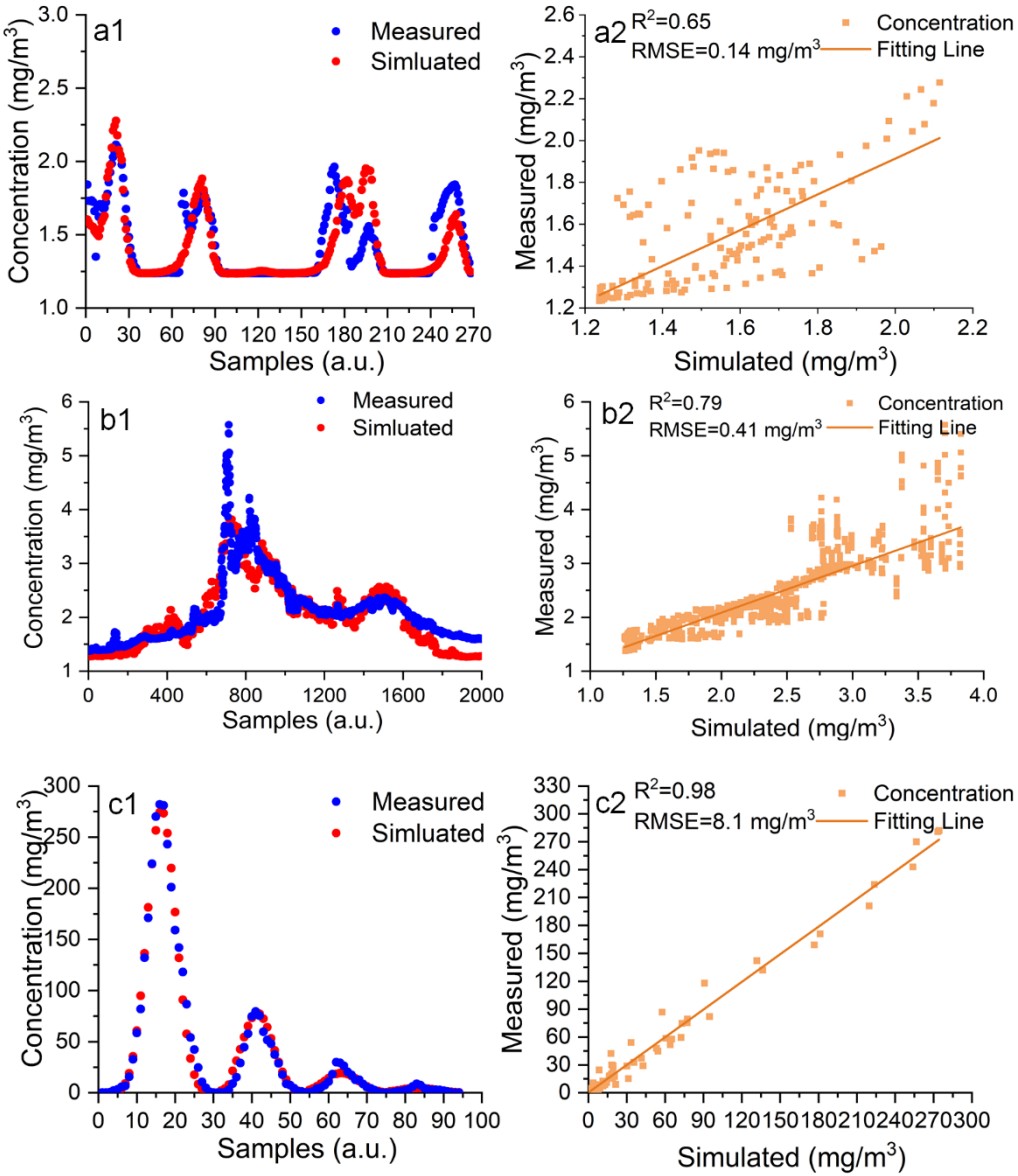

**Fig.10.** The simulated gases diffusions based on retrieved parameters in control release experiments; a1 and a2 are comparisons between simulated diffusion and actual samples in Agrar Hauser; b1 and b2 are comparisons between simulated diffusion and actual samples in EPA control release; c1 and c2 are comparisons between simulated diffusion and actual samples in Prairie Grass experiment.

As shown in Fig.10, the gases diffusions simulated by GA-IPPF in the three control release experiments conform to logic. Simulated gases concentrations are in good agreement with actual samples (see Fig.10.a1, Fig.10.b1 and Fig.10.c1,), and each peak of the samples in control release experiments can be reconstructed. The correlations between simulated gases concentrations and actual samples are larger than 0.65, and the RMSE are within 2.7% (relative to the mean value of the selected samples' concentration).

In general, reconstructions of gases concentration based on both mobile-platform and UAV are worse than that based on in-situ network. Collected data by in-situ network is usually the mean value of a certain time, like 10 min in Prairie Grass emission experiment, which provides stabile inputs data for GA-IPPF, especially concentration samples. While the concentrations sampled by mobile-platform and UAV-based

AirCore experiments are instantaneous, which may be inaccurate and exist fluctuations in collections. The advantages of vehicle-based and UAV-based sample systems are flexibility, that is, they can freely acquire the distribution of gases around the target monitoring sources. In-situ network implement is complicated with a high cost, and the wind direction should be considered during deployment. But its high stability and accuracy can help us to quantify emission source. Therefore, environmental protection

departments can choose detection systems according to actual emission monitoring needs.

**5.Conclusion**

In this study, we present a quantified model for strong point emission source based on concentration sampling data, named GA-IPPF. During CoMet campaign in 2017, we successfully monitor methane emissions from a ventilation shaft in Pniówek coal mine through the concentration data measured by

505 UAV-based AirCore system. Results show that $CH_4$ emissions rate from ventilation shaft are not consistent even in a short time.

GA-IPPF can reconstruct the concentration dispersion around the point emission source, and the largest $R^2$ between the measured $CH_4$ concentration and the reconstructed concentration in the selected Flights around Pniówek coal mine can reach 0.99.

In Observing System Simulation Experiments (OSSE), we discussed the sensitivity analysis of different parameters setting on final retrieved emission rate by GA-IPPF. We demonstrate that GA-IPPF has self-adjust function to achieve an optimal solution on emission rate, which will reduce the requirements for hardware performance in actual emission quantification experiment.

We also tested the performance of GA-IPPF in three control release experiments with different

sampling devices, including vehicle-mounted in-situ system, UAV-based AirCore system and ground-based in-situ network observation, and the biases between retrieved emission rates and reported ones within 5.0 %.

In future, GA-IPPF has great potential in the point-source quantitation based on mobile concentration sampling system, which can help to renew and enrich the gases emission inventories on strong point

sources.

*Author contributions.* HC, XM, TS planned the campaign; TS, TA, HC, HM performed the measurements; TS and ZH analyzed the data; GH and TS wrote the manuscript draft; CC, HZ, and WG reviewed and edited the manuscript.

*Competing interests.* The authors declare that they have no conflict of interest.

*Acknowledgements.* This work was supported by the National Natural Science Foundation of China (Grant No.41971283, 41801261, 41827801, 41901274, 41801282, 42171464), the National Key Research and Development Program of China (2017YFC0212600), The Key Research and Development Project of Hubei Province (2021BCA216), the Open Research Fund of National Earth Observation Data Center (NODAOP2021005).

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
