# Peer review of "Retrieving CH4 Emissions from Coal sampled with UAV-based Aircore system by using GA-IPPF model"

_Atmospheric Chemistry and Physics, 2022_

## Referee Comment (RC1)

Overall – the method appears potentially impressive from the data presented here, but the quality of the paper is let down by some imprecise writing and concerns around cherry picking of data for presentation. It feels that it needs some sort of controlled release validation to be fully compelling as a go-to method of quantification. Although probably out of scope of this work (given that this is part of the CoMET special issue) this quantification method should be tested with data collected at blind control release experiments to ensure that it is truly capable of describing real emissions, and therefore able to correctly understand the "real" uncertainty rather than the mathematical uncertainty within these quantifications. As it stands, I feel the authors need to explain some parts in more detail to alleviate concerns and make some significant improvements to the quality of writing throughout.

It will be suitable for full publication once the problems are ironed out.

**Major thoughts**

The volatility of emissions from mining operations is not surprising given the variable conditions and operational mine venting. The aircore system is a well established method of data collection and it is good to see it being able to capture the emissions nice and clearly. I have concerns about the data quality though for being able to fully resolve the mass balance with any certainty. In Figure 3a, the plume is only intercepted once along a single transect and is not fully bounded with background air to the South. In Fig 3b, there appears to be two hotspots, with the main lofted plume not resolved to the West at all. This is one of the downsides of not measuring inflight and only being able to measure post-flight. Whilst the uncertainties of the GA-IPPF are low, it needs to be stated that we do not know the error in the accuracy of the method as we have no pre-defined truth to compare to.

The issue of quantification of methane from the energy sector has the potential to become potentially political and legal – with the concept of emissions levies or preferred contracts based on quantified methane emissions being used as a legal instrument. It is therefore of upmost importance that the authors of novel methodologies for quantification such as those shown here are aware of how their methods may be utilised in the future, potentially by 3rd party commercial companies, and ensure that any issues are fully declared (e.g. why are only 2 flights used to verify the method, were there occasions when this method failed, and why? There are 15 flights stated on the experiments section). They should also check that all references to accuracy are truly discussing accuracy (closeness to a true value) – and not precision (closeness to another measurement or model).

If this method is suitable, I think a comment section needs to be added discussing other sorts of data that the team would expect this method to work with. I'm assuming that on-board drone measurements downwind from landfill / industrial sites would be a good option, would mobile measurements from vehicles potentially work?

**Minor**

I am far from a professional proof-reader, and would recommend that this manuscript is looked over by a proofer after corrections as there are numerous tense and grammar issues that need resolving to make the paper read as desired.

L20: There are plenty of monitoring methods – but very little verified quantification methods suitable for coal mines.

L34: Grammar. Release is concerning.

L38: Why are BU only useful for strong sources?

L42: This seems to be a common misuse / expectation of a BU inventory. They are not intended to be able to capture variable emissions, but are a statistical average expectation. It is only worth considering inventories compared to spot measurements if there is some valid statistical analysis (either temporal, or numbers of sites)

L44: What improvements have suddenly made this possible?

L46: Tense: should be is capable of obtaining.

L59: But most aircraft equipped with any CH4 sensor are able to achieve ppb precision and are perfectly capable of measuring downwind flight plans from coal mines. Many aircraft are capable of this such as Scientific Aviation etc...

L62: What about ground based eddy covariance?

General: to L70. This section feels unnecessarily negative about the capabilities of the rest of the scientific community with regards to being able to measure emissions from coal mines. There are several methods discussed here that I would envisage perfectly capable of making precise measurements that could enable a quantification of emission estimate.

L73: What does high applicability mean in this sense?

L74: How will it have less uncertainty if the inputs still have the same uncertainties attached to the actual measurements? The discussion to L83 makes it seem that these important atmospheric parameters can be discarded in favour of a set of model parameters? From experience at controlled release experiments small changes in wind direction and other meteorlogical conditions can have a dramatic effect on the plume behaviour.

L106: Is the accuracy only 20ppb using a G2401-m? This is concerning, why is it so large? What is the location accuracy of the sampling, how much does the sample bleed into itself over the course of a flight? Is the mixing in the aircore linear across flight time?

L117: correlation not connection? How good a correlation?

L135: I'm not 100% sure what alpha represents, please clarify (with a reference if possible)

Approx. L160. Is there prioritization in the fitting process? Are some variables given priority due to their certainty? General question about the process, if the inputs are very close to the outputs, then I presume that the standard gaussian plume quantification would also be very close?

L190: If these are spiral patterns, then the figure doesn't make this very clear. Can these be replotted to make it clear if that is the case?

Para 218: This paragraph needs tidying up, it is unclear as it currently reads and has numerous typos. The reconstruction is impressive – especially with two peaks in Flight 15 so it is important that this section is as simple and clear as possible.

L250: If the Gaussian plume doesn't account for background, has the dataset been adjusted to make that correction?

Approx L265: I don't know if this is possible, but it would be incredibly helpful if there was some sort of visualization of the CH4 atmospheric concentration for each of the methods of quantification (e.g. what does the plume look like in 2-D so that the variability can be understood). This would be most helpful for where there is significant differences between the methods to show what the plume visualisation looks like in each of the quantifications.

L330 (approx.) General discussion of measurement stability – the variation in wind parameters, disturbance in the airflow from the drone, changes in plume behaviour are all potential problems for quantification. Are the random errors used here sufficient to capture the potential uncertainty and can they be justified with reference to uncertainty from other studies?

L381. The claim that this result would guarantee emission calculation accuracy to better than 99.2% seems very over confident. This type of statement brings me back to the knowledge that methane emissions may be used as legal instruments in the near future and claims of such accuracy when not compared to a blind control is not acceptable. There needs to be a clear rethink of the use of accuracy, error and precision throughout so that it is clear what is being compared. As there is no "known" value, no claim on accuracy can be made.

L401. The conclusions feel very generic and non-specific in large parts to the work shown here, they should be reconsidered with the value added of GA-IPPF in mind, rather than generalities of coal emissions. It is good to see that the model is being considered for other uses too, is there any possibility that this could be expanded on in the main text to demonstrate how this would be done for vehicles?

---

## Referee Comment (RC2)

**Retrieving CH4 Emissions from Coal sampled with UAV-based Aircore system by using GA-IPPF model**

Tianqi Shi[1], Zeyu Han[2], Ge Han [3],Xin Ma[1*],Huilin Chen[4,5*], Truls Andersen[5],

Huiqin Mao[6],Cuihong Chen[6],Haowei Zhang[1],Wei Gong[1,7]

[1] State Key Laboratory of Information Engineering in Surveying, Mapping and Remote Sensing, Wuhan University, Luoyu Road No.129, Wuhan 430079, China;

[2]School of Mathematics and Statistics, Wuhan University, Luoyu Road No.129, Wuhan 430079, China;

[3] School of Remote Sensing and Information Engineering, Wuhan University, Luoyu Road No.129, Wuhan 430079, China;

[4]Joint International Research Laboratory of Atmospheric and Earth System Sciences, School of Atmospheric Sciences, Nanjing University, Nanjing, China;

[5]Centre for Isotope Research, Energy and Sustainability Institute Groningen (ESRIG), University of Groningen, Groningen, Netherlands;

[6]Ministry of Ecology and Environment Center for Satellite Application on Ecology and Environment, Beijing, China;

[7] Electronic Information School, Wuhan University, Luoyu Road No.129, Wuhan 430079,China;

*Correspondence to*: Xin Ma(maxinwhu@whu.edu.cn); Huilin Chen (huilin.chen@rug.nl)

**Abstract**

Quantifying CH4 emissions from coal mines has large uncertainty owing to the lack of effective monitoring methods. In this study, we developed a genetic algorithm–interior point penalty function (GA-IPPF) model to calculate the emission rate of large point sources of CH4. This model can provide the detailed optimized dispersion parameters and has self-calibration characteristics that reduce the accuracy requirements for auxiliary data. We evaluate the influence of different parameters on the retrieved CH4-emission rate by the GA-IPPF, including the uncertainty of CH4 concentration measurements, number of CH4 measurements, and meteorological data. Furthermore, based on the atmospheric CH4 concentration measurements downwind of a Pniówek coal-mine ventilation shafts from a UAV-based AirCore system and the GA-IPPF model, we retrieve the CH4-emission rates from the Pniówek coal (Silesia coal mining region mine, Poland) ventilation shaft. Results show that the CH4-emission rates are variable even in a single day, ranging from 5.6±0.2 kt/year to 12.4±0.6 kt/year on August 18, 2017 and from 3.0±0.3 kt/year to 12.7±0.5 kt/year on August 21, 2017. The combination of the flexible UAV-based AirCore CH4 measurements and the robust GA-IPPF model provides a effective means of quantifying CH4 emissions from coal mines.

**1.Introduction**

The release of $CH_4$ into the atmosphere during coal mining is very concerning because it contributes to increased atmospheric concentration of $CH_4$, one of the most important greenhouse gases and waste resources(Cardoso-Saldana and Allen 2020; Zhang et al. 2020). However, $CH_4$ emissions during coal mining are not always stable owing to collection or manufacturing processes, weather fluctuations, terrain effects(Nathan et al. 2015b). Bottom-up inventories could provide us with approximate information on $CH_4$-emission rates from strong sources. However, inventories cannot serve as a standard for formulating policies to reduce anthropic $CH_4$ emission because of their low temporal resolution and uncertainty. The temporal resolution and accuracy of bottom-up inventory are too low to obtain emission information instantaneously(Liu et al. 2020; Pan et al. 2021). Based on the measurements of $CH_4$

concentration around the emission source, developing a fast retrieval model to obtain emission intensity is an acceptable method. With the development of remote-sensing technologies, the $CH_4$ emission rate has become possible to quantify based on $CH_4$ concentration samples or measurement.

Greenhouse gases observing satellite and TROPOspheric Monitoring Instrument could obtain the column concentration of $CH_4$ ($XCH_4$, ppb) with a spatial resolution of 10 km×10 km and 5 km×7.5 km.

The regional $CH_4$ flux could be retrieved by assimilating the measured $XCH_4$ into an atmospheric dispersion model(Feng et al. 2016; Tu et al. 2022). PRISMA hyperspectral imaging satellite and GHGsat could detect increased $CH_4$ caused by strong emission sources with high spatial resolution, and the comprehensive $CH_4$ emission could be quantified by integrated mass enhancement or cross-sectional flux method(Guanter et al. 2021; Varon et al. 2020). However, $CH_4$ emission from coal is not constant

*Poor English and not clear - you mean that the orbital pattern of the satellite means that they do not pass over the same area twice within the same day.*

even in a short time, and the spatial and temporal resolutions of satellites are not allowed to repeat the quantification of $CH_4$ emission from coals in the same day(Schneising et al. 2020; Varon et al. 2019). An airborne platform could fly in low altitude to improve the acquisition of $CH_4$ concentration(Elder et al.

2020; Wolff et al. 2021a) and estimate $CH_4$ emission from strong sources by cross-sectional flux method or Gaussian dispersion method. However, it has strict requirements for the flight track (downwind

*I presume you mean fixed location. This is also dependent on wind direction, so the emission will not be sampled all of the time, unless monitoring is inside of the source.*

direction) and amount of measured $CH_4$ concentration data. Most ground-based sensors have the advantage to sample the concentration around the source continuously, but it could collect data only near the surface or could only measure column concentration(Caulton et al. 2017; Robertson et al. 2017; Zhou et al. 2021), which are insufficient to generate the distribution characteristic of the emission source.

Ground-based different-absorption LIDAR could obtain the $CH_4$ profile concentration in different altitudes, which is suitable as the input of the emission-retrieval model(Shi et al. 2020a), but it has high requirements in terms of performance of hardware and system stability(Shi et al. 2020b). An unmanned aerial vehicle (UAV) could reach any location rapidly around the $CH_4$ sources, which could sample $CH_4$

concentration with location information(Iwaszenko et al. 2021; Nathan et al. 2015b), when equipped with an in-situ gas sensor. It could also acquire the characteristic of distribution with adequate data, which is beneficial to retrieving the emission rate.

Mass-balance method has been applied in determining $CH_4$ emissions based on UAV-based samples(Allen et al. 2018). Emission rates calculated by this method contain large uncertainty because the main kernel is Kriging interpolation(Nathan et al. 2015a), which causes obvious uncertainty in representing the actual feature of diffusion. The Gaussian dispersion model has also been applied in retrieving gas emission from strong sources(Shah et al. 2019)[30], and is used to quantify $CH_4$ emission from a coal-mine ventilation shaft in this study. However, existing emission-retrieval models need priori information on diffusion parameters to retrieve the emission rate. Moreover, the accuracy of measurements of auxiliary meteorological data influences the result of $CH_4$-emission calculation.

Therefore, we develop herein a model to overcome these shortcomings. Our model could calculate the diffusion parameters without prior information and reduce the impact of meteorological data on the calculated $CH_4$-emission rate.

In the present study, we collected $CH_4$ concentration around a coal-mine ventilation shaft by using

*data*

*Accuracy of what? AirCore only collects the sample so presumably GPS location X.Y.Z co-ordinates at high frequency.*

UAV-based active AirCore system with high accuracy(Andersen et al. 2018) for a total of seven Flights.

Then, a $CH_4$ emission-retrieval model based on genetic algorithm (GA) combined with interior point- penalty function (IPFF) is presented. GA-IPPF could help us obtain detailed information on emission intensity and diffusion parameters. Finally, the performance of GA-IPPF model is compared with other quantification methods for $CH_4$ emission.

**2.Data and methods**

*2.1.Active AirCore System*

The active AirCore system comprises a ~50 m coiled stainless-steel tube that works in conjunction with a micropump and a small pinhole orifice (45 $\mu m$) to sample air along the trajectory of a drone. As long as the pressure downstream of the orifice is more than half of that of the upstream (ambient) pressure, a critical flow through the orifice is obtained. This finding means that the flow rate depends only on two variables, namely, the air temperature and the upstream (ambient) pressure, both of which are monitored during the flight. After obtaining the air sample, the sample is analyzed on a cavity ring down

Need to be clear that the AirCore sample is removed from the drone and replayed through the Picarro on the ground

Spectrometer model G2401-m for $CO_2$, $CH_4$, and CO. For $CH_4$, the accuracy of samples is ±0.02 parts

Samples are not accurate, measurements are, but I presume that the reference here is to precision not accuracy, as you need instrument calibration to measure accuracy.

[revised manuscript text omitted]

The results are treated as the final retrieved values of the concerned parameters when the R reaches the maximum. We use the "fmincon (interior point)" toolbox in MATLAB 2020a to implement the IPPF model.

[Figure]

**Figure1.** Flow chart of Genetic Algorithm and IPPF model.

*2.4.Measurement Site*

This section needs to go before the model section as the location is mentioned. You should also make it clear what type of source this is as this will make a big difference to the model. Is a a large open coal pit, or emission from a deep mine shaft or vent, or is it a combination of both.

The Pniówek coal mine (49.975° N, 18.735° E) is a large mine in south Poland in Pniówek, Silesian

Voivodeship, which is 190 km southwest of the capital Warsaw, see Figure2. It has a large coal reserve estimated to be about 101.3 million tons. Its coal production is about 5.16 million tons per year.

[Figure]

**Figure2.** The Pniówek coal mine

Caption needs more explanation. What actually is the zoomed map showing? Where is the mine? Is pink the mine buidlings? is it deep mine with shaft or open pit?

**3. Results**

*3.1. Validation of performance of GA-IPPF model through simulations*

First, the dispersion of CH$_4$ emission from a coal is simulated by equation 1, and the dispersion parameters are shown in Table 1. To make the simulations close to the actual measurement scenarios, random errors are added to the CH$_4$ concentration samples (5%), wind speed (± 0.3 m/s), and wind direction (± 20°). The spatial resolution of the supposed samples is 10 m, and 70 samples are selected from the simulated dispersion to represent the data acquired by the UAV-based AirCore. Then, the concerned parameters are retrieved by the GA-IPPF method. The input parameters include hypothetical wind speed, wind direction, and 70 samples, as shown in Figure3. Simulations are repeated 10 000 times, and the average values of the corresponding parameters are treated as the "Retrieved" results in Table 1.

[Figure]

**Figure3.** The determined 70 CH$_4$ samples in simulations

Table 1.Set parameters in dispersion simulation and the retrieved results

| Parameters | Actual | Retrieved |
|---|---|---|
| Emission intensity (g/s) | 300 | 300.5±0.01 |
| Wind speed (m/s) | 3 | 3±0.01 |
| Wind direction (°) | 90 | 90±0.01 |
| a | 0.11 | 0.13±0.02 |
| B | 0.9 | 0.9±0.02 |
| c | 0.1 | 0.12±0.01 |
| d | 0.82 | 0.8±0.01 |
| B (ppb) | 1900 | 1900±3 |
| Emission height (m) | 20 | 19.7±1.2 |
| α | 0.9 | 0.89±0.03 |

"Actual" means the set values of parameters, and "Retrieved" means the average values of parameters

   retrieved by GA-IPPF model through 10 000 times of simulation.

As shown in Table 1, q retrieved by GA-IPPF has only 0.17% bias compared with the set values.

Emission height only has 0.3 m bias to set one, and uncertainty is only 0.6% to 20 m. Other retrieved parameters also show high consistency with the original settings.

*3.2. Stability analyses*

The necessary input parameters in GA-IPPF contain meteorological data (wind speed and direction), accuracy of CH$_4$ samples, and amount of CH$_4$ samples. In formula 1, wind speed nearly has a linear relationship with the emission estimation. Wind speed is also an important factor that determines atmospheric stability according to the Pasquill–Gifford method(Venkatram 1996) as it affects the diffusion parameters of $\sigma_y$ and $\sigma_z$. The coordinate is built according to the wind direction, which is defined

      something is missing here the plane coordinates of $CH_4$ samples. According to formulas 2 to 3, errors in wind-direction measurement lead to wrong $\sigma_y$ and $\sigma_z$ on each position of samples. $CH_4$ samples are the most important factors to determine the Gaussian diffusion. The accuracy of samples influences the judgment of "fitness"

in the GA process. More samples collected in different positions help rebuild the spatial-distribution characteristics of the plume because it provides larger possibility for fitting process in IPPF and helps determine the optimum solution. To evaluate the influence of errors in the measurements of necessary parameters on the final retrieved results, the same settings in Table 1 are used as actual results. The performance of the GA-IPPF model with additional random errors in each parameter are simulated

10 000 times, as shown in Figure4.

[Figure]

**Figure4.** Influence of accuracy of parameters on final results. Baseline represents the set emission rate of $CH_4$, 300 g/s: (a) wind speed, with additional error ranging within 0.2–2 m/s and an interval of 0.1

m/s, (b) wind direction, with additional error ranging within 5°–40° and an interval of 5 °, (c) accuracy of $CH_4$ samples, with additional error ranging within 0.5%–5.0% and an interval of 0.5%, and (d) amount of $CH_4$ samples, randomly selected as 20–70 among the defined 70 samples.

      Need to make clear earlier that q = emissions in g/s

In Figure 4 (a), the mean value of q retrieved by GA-IPPF has nearly the same with the baseline if the error in wind speed is less than 0.4 m/s and the maximum bias to the baseline is 16.3 g/s. Fluctuation of q occurs obviously if error in wind speed exceeds 0.4 m/s. The standard errors of q are positively correlated with the values of errors in wind speed, indicating that the accuracy of wind-speed measurements largely influences the stability of the GA-IPPF model. This model has a self-adjustment function for wind speed; for example, when the oral wind speed is 3 m/s, the maximum standard error of q is only 8.5 g/s (3.5% to the 300 g/s) when the additional error of wind is 2 m/s (66.7% to 3 m/s).

The retrieved q shows less sensitivity on errors in wind direction, see Figure4(b). When errors in the wind direction are 5° to 40°, all biases of q are within 1.1 g/s and the standard errors are around 2.3 g/s.

Wind direction determines the spatial location of the sampling point, and wrong location information leads to distinct errors in emission estimation. GA-IPPF shows highly accurate ability in wind direction

This implies that there could be big errors in q if the source
is not a point source, but unequally distributed over an area,
which could be the case with a farm or landfill site emission.

to obtain the global optimum solution.

Sampling accuracy has small impact on the retrieved q within different settings in CH$_4$ samples'

accuracy, see Figure4(c). Standard deviation is positively correlated with errors in CH$_4$ measurements.

The standard deviation is 3.1 g/s when the measurement error reaches 5%. Notably, the uncertainty of

CH$_4$ samples measured by UAV-based AirCore system is far less than 5%. The AirCore system could acquire more than 70 CH$_4$ samples in actual feasible measurements, thereby guaranteeing the accuracy of the retrieved CH$_4$ emission by coal to exceed 99.2%.

The number of measurement points obviously influences the final accuracy of q by the GA-IPPF model, see Figure4(d). It has 5.9 g/s bias to 300 g/s when n is 20. The accuracy of q and the standard error are negatively correlated with n, which provides the number of judging criteria for the fitting process in the retrieval model. Hence, n directly influences the retrieved results. The AirCore system has the advantage of continuous sampling during flight, which integrates the atmospheric signals along the flight path and helps reduce the uncertainty of the retrieved q. On the other hand, the smoothing of the atmospheric signal also reduces the spatial resolution of the measurements, which needs to be considered during the optimization[30].

IPPF can suitably solve the problem of inequality constraints, and the calculated solution guarantees the calculated parameters to be within the feasible region. In this section, GA-IPPF model performance and its adjustments on the concerned four input parameters were discussed.

*3.3. Actual experiments*

Fifteen active AirCore flights around Pniówek coal mine are collected successfully on August 18, 2017

and August 21, 2017. The sample data in Flight 8 (18/8/2017) and Flight 15 (21/8/2017) are used to evaluate the GA-IPPF model, as shown in Figure5.

[Figure]

**Figure5.** Samples of CH$_4$ in two Flights: (a) Flight 6 and (b)Flight 15.

In Flight 5, the AirCore system collects CH$_4$ around the coal spirally from 0 m to 98 m, for a total of

376 samples, and the measurement period is 7 min, ranging from 1980.1 ppb to 49 113.9 ppb. In Flight

15, the AirCore system collects a total of 400 samples, and the measurement period is 9 min, ranging from 2131.7 ppb to 57 265.3 ppb. Both Flights show high spatial variability in CH$_4$ exhaust from coal mine. Subsequently, we input the wind speed, wind direction, location information, and CH$_4$ samples collected from Flights into the GA-IPPF model. To express the final retrieved emission (Q) in g/s, the dry-air mixing ratio of CH$_4$ C (ppb) is transformed into mass concentration m (g/m$^3$) as follows:

$$m = C \cdot \frac{M_{CH4}}{M_{Air}} \tag{8}$$

$M_{CH4}$ is the molar mass of $CH_4$, $M_{air}$ is the molar mass of air.

The retrieved results are shown in Table 2. Notably, the emission height in Flight 15 is larger than that
of Flight 6, which may be caused by the difference in thermal energy and vertical wind speed of the two
flights. The background concentrations of $CH_4$ are 2001.3 and 2002.1 ppb in Flights 6 and 15,
respectively, which show little difference. The dates of the two Flights are very close, so the background
concentration of $CH_4$ in two days have nearly the same seasonal characteristic. The exhaust gases of coal
mine are emitted through the stack with effective emission heights of 5.7 and 3.64 m, respectively.

Why is the calculated emission height an order of magnitude higher in the model results?

To evaluate the rationality of the retrieved results, these parameters are used to simulate $CH_4$ diffusion
from the Pniówek coal mine according to equation 1. The comparison between simulated $CH_4$
concentration and actual samples in the same locations is shown in Figure6.

                    Table 2. Results calculated by GA-IPFF model

[revised manuscript text omitted]

NLSF and the combination of NLSF with Gaussian diffusion model are also extensively used for point- source emission retrieval (Wolff et al. 2021b; Zheng et al. 2020). In this study, NLSF is used to estimate

Q in each Flight by fitting the unknown parameters in equation 1, and the uncertainty of the retrieved Q

is evaluated with formula 6.

Andersen et al. also developed an inverse Gaussian approach to quantifying $CH_4$ emissions from coal mine based on the same Flights (Andersen et al. 2021). First, the Gaussian dispersion is built as

$$C(x, y, z) = \frac{q}{2\pi u \sigma_y \sigma_z \cos(\theta)} \exp(\frac{-(y)^2}{2\sigma_y^2}) \left\{ \exp(\frac{-(z-H)^2}{2\sigma_z^2}) + \alpha \cdot \exp(\frac{-(z+H)^2}{2\sigma_z^2}) \right\} \tag{12}$$

Where $\theta$ is the angle between the wind direction and the perpendicular line of the flight trajectory. This model does not contain the item of background of $CH_4$. Furthermore, $\sigma_y$ and $\sigma_z$ are treated as certain values in equation 8. Then, q and the related parameters are retrieved by "fmincon optimizer" in

MATLAB (detailed settings are found in Andersen et al.). The $CH_4$ emissions in each Flight, as evaluated by Andersen et al., are presented in this section.

Facility-emission data and hourly $CH_4$ emission from shaft are calculated by measuring raw $CH_4$

concentration and air flux through the shafts. The $CH_4$ emission rates estimated using hourly facility- emission data for 21 August 2017 and 21 August 2017 are 14.4 ±4.9 and 8.2 ±2.9 kt/year, respectively, as shown in Figure7.

[Figure]

Why is the inventory emission lower for flight 8? Seems to be consistent across the second day.

**Figure7.** Quantified $CH_4$ emission by different methods based on the collected data: (a) August 18, 2017

and (b) August 21, 2017. The results of $CH_4$ emission rate calculated by Mass balance and Inverse

Gaussian refer to Andersen et al[30].

As shown in Figure7, Flights 4, 6, and 8 are measured on 18 August 2017, whereas Flights 12, 14, 15,

17, and 18 are measured on 21 August 2017. Figure7(a) shows that the $CH_4$-emission rates calculated by mass balance are smaller than the inventory estimation in all Flights. In Flight 8, q retrieved by mass balance is extremely lower than those quantified by other methods, whereas q retrieved by GA-IPPF

model (12.4±0.6 kt/year) shows only a slight difference from the inventory. As shown in Figure 7(b),

$CH_4$ emissions retrieved by mass balance, inverse Gaussian, and GA-IPPF model are overestimated compared with the inventory in Flight 12. Mass balance and inverse Gaussian method also show obviously underestimated q in Flight 17. Estimations of retrieved $CH_4$ emission in Flight 18 show consistency among methods of mass balance, GA-IPPF, and inverse Gaussian. The $CH_4$-emission rate of coal generally has significant variability in each measurement, even on the same day. Mass balance is very sensitive to the size settings of grids, and different height and length settings can affect the concentration distribution across the cross-section. NLSF has a high-accuracy requirement for wind measurements, and errors on these measurements have a linear influence on the final emission estimation.

Notably, the standard errors of q quantified by GA-IPPF always are the least among these methods, indicating the stability of our developed model.

*4.2 Application of Reanalysis meteorological database in GA-IPFF model*

Wind speed and wind direction acquired by the radiosonde or weather station are two main parameters in GA-IPPF. However, additional sensors are bound to increase the cost and difficulty during actual $CH_4$- emission measurements. To explore the possibility of weather reanalysis data instead of actual wind measurement by sensors, we use 10 m U and V wind components from the ERA5 meteorological reanalysis database (spatial resolution is 0.1°×0.1°, and temporal resolution is 1 h) developed by the

European Centre for Medium-range Weather Forecast(Hersbach et al. 2020) to evaluate GA-IPPF model.

However, the wind directions from ERA obviously differ from the actual measurements during the

Flights. Hence, we determine the wind direction by using the $CH_4$ samples, for example, the line between the shaft and the location of the maximum value of samples in the same heights is treated as the downwind direction, whose uncertainty is set as 50°. Wind speed from ERA is used for the $CH_4$-emission calculation, uncertainty is supposed as 2 m/s. Even oral wind speed and direction obviously differ between the two sources; however, the GA-IPPF model adjusts them into reasonable ranges. The results of q during all Flights retrieved by two meteorological data sources have been evaluated, as shown in

Table 3.

Table 3. Retrieved CH$_4$ emission by ERA meteorological data

| Flights | Actual | ERA |
|---------|--------|-----|
| 4 | 5.6±0.2 | 6.0±0.3 |
| 6 | 6.1±0.3 | 6.4±0.5 |
| 8 | 12.4±0.6 | 14.4±0.9 |
| 12 | 12.7±0.5 | 13.2±0.7 |
| 14 | 6.4±0.3 | 5.0±0.4 |
| 15 | 8.4±0.5 | 7.8±0.5 |
| 17 | 3.0±0.3 | 3.4±0.5 |
| 18 | 6.5±0.4 | 7.0±0.5 |

Table 3 shows that the values of quantified q between two meteorological sources are within 20% in
the same Flight. The standard errors of q retrieved by the ERA5 database are larger than those from actual
measurements, which depends on the accuracy of the reanalysis of wind speed and wind direction. Thus,
it reduces the necessity of additional equipment except for the AirCore system and the complexity of this
program.
To explore the reason that the acceptable difference of calculated methane emission rate by the two
sources of meteorological data. The concerned parameters in Flight 6 and Flight 15 calculated based on
ERA5 meteorological data were [are] presented in Table 4.

Table 4. Results calculated ERA5 meteorological data

| Parameters | Flight 6 | Flight 15 |
|-----------|----------|-----------|
| Oral wind speed (m/s) | 2.6 | 4.1 |
| Oral wind direction (°) | 300 | 120 |
| Emission intensity (kt/year) | 6.4±0.5 | 7.8±0.6 |
| Wind speed (m/s) | 2.99 | 4.52 |
| Wind direction (°) | 349.4° | 128.1 |
| a | 0.28 | 0.18 |
| b | 0.90 | 0.93 |
| c | 0.01 | 0.13 |
| d | 1.26 | 0.84 |
| B (mg/m$^3$) | 1.56 | 1.57 |
| Emission height (m) | 60.2 | 36.0 |
| Reflection index | 0.80 | 0.71 |

The oral wind speed and wind direction in Table 4 are obviously different from those in Table 2.
However, the calculated wind directions are nearly the same based on the two sources of meteorological
data. Diffusion parameters and emission height in Table 8 also show less difference in two Tables (Table
2 and Table 4). It is worth nothing that the wind speed and reflection index would be adjusted to reach
the global solution by GA-IPPF model, which leads to little bias for the emission rate of CH$_4$ in Table 3.

[Figure]

**Figure8.** Comparison between the measured samples and the simulated ones based on the ERA5 meteorological data: (a). Flight 6 and (d) Flight 15. The difference of simulated CH4 concentration and actual measured ones:(b) Flight 6 and (e)Flight 15. Correlation Analysis: (c) Flight 6 and (f)Flight 15.

The simulated concentration of $CH_4$ in Flight 6 and Flight 15 calculated by parameters in Table 4 are shown in Fig.8. In Fig.8(a), the consistency between actual samples and simulated ones is slight lower than that in Fig.6(a), $D_c$ is ranging from -2.4 to 4.3 mg/m$^3$, which is an acceptable bias as only 6 points exceed 2.3 mg/m$^3$. $R^2$ (0.98) of measured samples and simulated ones is almost same to that in Fig.6(c), while RMSE is nearly three times than that in Fig.6(c). In Fig.8(d), the tendency of simulated $CH_4$ concentration is similar to Fig.6(d). $D_c$ is ranging from -11.9 to 11.6 mg/m$^3$, which is nearly same as the result in Fig.6 (e), it worth nothing that $D_c$ simulated by ERA meteorological data is slight larger on samples (NO.1 to 20) compared with that in Fig.6 (e). The $R^2$ and RMSE in Fig. 8(f) indicate that the retrieved results using ERA data are less accurate than that using actual measured meteorological data. In summary, though we set large uncertainties in ERA meteorological data, GA-IPPF can still guarantee reasonable and adequate accuracy for the retrieved emission rate and diffusion parameters.

**5.Conclusion**

$CH_4$ emissions from coal are inconsistent even with short time differences. They usually show a large difference for different mining volumes and types. Enhancement in $CH_4$ by the emission source is much larger than the background concentration, and the distribution of leak gas shows an obvious spatial difference. Hence, the retrieval time needs to be shortened for each emission measurement. AirCore has high portability and flexibility to measure $CH_4$ concentration around emission sources, accompanied by the GA-IPPF model, which is acceptable to calculate $CH_4$ emission from coals or other point sources. This program can help improve the accuracy of estimating $CH_4$ emission from coals, especially developed countries that even lack no inventories of gas emission. It can also help governments evaluate the fugitive $CH_4$-emission rate during mining and formulate policies to promote the innovation of mining equipment and technology. The recommended program is appropriate for quantifying local sources based on the advantage of hardware and retrieval model. The UAV-based AirCore system helps rebuild the diffusion of $CH_4$ with flexibility and high sampling rate. The GA model could restrict the calculated emission details within a reasonable range. Therefore, this program has great potential application in the point-source quantitation of other gases, such as $CO_2$, $SO_2$, etc. *Should include a recommendation of how many Air Core retrievals from a single point source are required to mininmise the errors on model results*

[revised manuscript text omitted]

---

## Author Comment (AC1)

Overall – the method appears potentially impressive from the data presented here, but the quality of the paper is let down by some imprecise writing and concerns around cherry picking of data for presentation. It feels that it needs some sort of controlled release validation to be fully compelling as a go-to method of quantification. Although probably out of scope of this work (given that this is part of the CoMET special issue) this quantification method should be tested with data collected at blind control release experiments to ensure that it is truly capable of describing real emissions, and therefore able to correctly understand the "real" uncertainty rather than the mathematical uncertainty within these quantifications. As it stands, I feel the authors need to explain some parts in more detail to alleviate concerns and make some significant improvements to the quality of writing throughout.

It will be suitable for full publication once the problems are ironed out.

Dear reviewer, thank you for all your kindly comments, which are very helpful in improving the quality of our manuscript. We think it is very reasonable to use the measured quantitative data to verify our proposed method, which can help readers to have more confidence in the presented calculation results. However, it is difficult to conduct control experiments for ventilation shafts in methane mines. Therefore, we use similar controllable experiments to verify our method. In our latest modified manuscript, we evaluate the performance of GA-IPPF in three release-controlled experiments: the vehicle-mounted $CH_4$ dataset of EPA, the $CH_4$ release dataset with Aircore system, and the $SO_2$ dataset measured by ground-based in-situ network.

The results of the above experiments showed the high accuracy of GA-IPPF, and the reconstructed methane diffusion results are consistent with actual measured ones. These control experiments can not only be used as a verification part, but also as a proof of whether GA-IPPF is applicable to the quantification of emission sources by other measurement systems.

As you mentioned, since this is the COMET special issue, the main purpose of our paper is to highlight the performance of GA-IPFF in assessing methane emissions in COMET. Thus, the evaluation of release-controlled experiments are presented in Discussion section.

Finally, we are very grateful for your approval of our manuscript, and we have organized and answered your comments one by one. Words in blue color are your comments, and our responses are in black color words. Detailed responses are as follows:

**Major thoughts**

1.The volatility of emissions from mining operations is not surprising given the variable conditions and operational mine venting. The aircore system is a well established method of data collection and it is good to see it being able to capture the emissions nice and clearly. I have concerns about the data quality though for being able to fully resolve the mass balance with any certainty. In Figure 3a, the plume is only intercepted once along a single transect and is not fully bounded with background air to the South. In Fig 3b, there appears to be two hotspots, with the main lofted plume not resolved to the West at all. This is one of the downsides of not measuring inflight and only being able to measure post-flight. Whilst the uncertainties of the GA-IPPF are low, it needs

to be stated that we do not know the error in the accuracy of the method as we have no pre-defined truth to compare to.

Thank you for this suggestion. First, when we used UAV-based Aircore system to collect the diffusion of $CH_4$ leakage, at a certain height, the UAV will fly laterally to ensure that Aircore system would detect larger value in the track. Because the high concentration can represent the diffusion characteristics of the emission source. GA-IPPF would ensure that a global optimal solution of unknow parameters can be obtained in Gaussian dispersion model. Therefore, theoretically, even if the maximum value or the entire cross-section is not obtained, GA-IPPF will also achieve optimal solution exactly. To verify this view, we performed a set of simulation experiments to compare the final inversion results based on fully- coverage and half-coverage in each cross section.

In this section, we simulated spatial distributions of atmospheric $CH_4$ enhancements due to emissions of a strong point source. The configuration of simulations is as follow. Uncertainty of wind speed is 0.3 m/s, and uncertainty of wind direction is 30°. The concentration accuracy of sampling is 0.5 %. Other parameters are showed in Table.R1.

Table.R1. Parameters and retrieved results by two modes of sampling

| Parameters | Actual | Full coverage (Fig.1a) | Half coverage (Fig.1 b) |
| --- | --- | --- | --- |
| Wind speed (m/s) | 2.5 | 2.47 | 2.45 |
| Wind direction (°) | 90 | 90.1 | 89.7 |
| Emission rate (kg/s) | 300 | 298.3 | 304.1 |
| a | 0.11 | 0.11 | 0.104 |
| b | 0.93 | 0.92 | 0.931 |
| c | 0.1 | 0.099 | 0.103 |
| d | 0.93 | 0.925 | 0.933 |
| H (m) | 20 | 20.2 | 20.7 |
| B (ppb) | 2000 | 2002.3 | 2003.2 |

As is shown in Fig.R1 and Table R1, the retrieved emission rate bias is 0.6 % to the set emission of 300 kg/s in full coverage mode, while 0.13 % to 300 kg/s in half coverage mode, therefore, we regard sampling mode in Figure 3a has little impact on the final result.

If the mass balance method is used to quantify methane emission, the incomplete sampling of hot spots in a cross-section will definitely affect the final retrieved emission rate. The incomplete sampling of hot spots means that the number of molecules in the selected cross-section is smaller than the actual one, resulting in the underestimate of the final quantitative results. The unknown sampled concentration is indeed a major drawback during the flight. If the mass balance method is used in actual experiment, the long-distance sampling shall be conducted in the direction perpendicular to the wind direction of the cross section as far as possible to ensure the integrity of the hot spot collection.

The two hot spots you nonmentioned in Figure 3b, is caused by the phenomenon that according to Gaussian diffusion, at a certain height, the closer the spatial position to the downwind direction, the greater the diffusion value. We also carried simulation experiments to show the vertical distribution at different heights. Fig.R2 showed that there are different hot spots in different heights, and Fig 3b in our manuscript shows the diffusion of $CH_4$ in heights of 16 m, 25 m and 40 m, and some samples of UAV-based Aircore system are collected closer to the downwind direction, hence, they show higher values in each altitude.

For "the main lofted plume not resolved to the West at all", it can refer to Fig.R1 and Table.R1. GA-IPPF would retrieve the unknown parameters according to different positions and the corresponding diffusion concentration. Half coverage mode would also ensure the high accuracy of the retrieved emission rate, see Table R1, the retrieved emission rate.

[Figure]

Fig.R1 Samples of two modes; a. full coverage; b. half coverage

[Figure]

Fig.R2 Simulated diffusions of $CH_4$ at different heights, a.50 m; b.100 m; c.150 m and d. 200 m.

As for the real experiments, we add three release-controlled experiments as verification. The three experiments are UAV-based, vehicle-mounted and in-situ net respectively. The emission sources are spatially sampled, and then the emission intensity is retrieved by the quantitative GA-IPPF. Among them, the emission deviation between the intensity obtained by GA-IPPF and the actual release value are 3.2 %, 5.0 % and 3.2%, respectively, which proved a good quantitative effect.

Detailed responses in our manuscript are as follows:

"**Emission Estimates in control release experiment**

To evaluate the performance of GA-IPPF in controlled release experiments, we quantified the emission rates in release experiment through different gases sample systems, including UAV-based Aircore system, mobile sampling system and ground-based in-situ network. Detailed introduction of the concerned release experiment are as follows:

**Agrar Hauser control release**

This $CH_4$ release experiment was conducted on Agrar Hauser field near Dübendorf, Switzerland. The controlled methane source was release from a 50 L high-pressure cylinder, the height of this artificial source is 1.5 m. meteorological information were acquired by 3D anemometers around the emission source. UAV-based sample systems used in these release experiments contains two sensors, including Quantum cascade laser spectrometer (QCLAS) and active Aircore. It carried a series active measurements from 23 February to 14 March 2020.There is no other $CH_4$ source around Agrar Hauser field and the topography is flat. In this section, active Aircore $CH_4$ samples on 12 march 2020 (312_01) are chosen to use GA-IPPF to quantify methane release rate.

**EPA methane control release**

Environmental Protection Agency (EPA),USA developed OTM 33A method to quantify oil and gas leakage based on mobile measurement platforms, which consist $CH_4$ in-situ sensor (G1301-fc cavity ring-down spectrometer (Picarro)), a collocated compact weather station and a Hemisphere Crescent R100 Series GPS system. The accuracy of in-situ sample is within ±5%, and in-situ sensor was implemented at height of 2.7 m based on vehicle. Weather station provides atmospheric temperature, pressure and humidity, as well as 3-D wind direction and wind speed. Simulated $CH_4$ leakage source was conducted using 99.9% methane high pressure cylinders as the gas supply. Total 20 experiments of control releases were published by EPA to evaluated OTM 33A.

**Prairie Grass emission experiment**

Prairie Grass emission experiment was mainly to evaluate the diffusion of $SO_2$ from point source under different meteorological circumstances [Barad, 1958], the height of emission source is 0.46 m, all in-situ sensors are set at heights of 1.5 m. $SO_2$ concentration sampled by in-situ network at radius of 50 m,100 m,200 m, 400 m and 800 m. Samples in R57 release (10-minute sampling periods), total 94, were selected to quantified $SO_2$ emission rate from release instrument. Reported emission rate of $SO_2$ in R57 is 101.5 g/s, samples collected in radius of 800 m was neglected in this discussion for amount of what is extreme less. Reported wind speed is 4.85±1 m/s, wind direction is 184±10°.

**Table 4 Performance of GA-IPPF model in different control release experiments**

| Database | Number | Gas | Release rates (g/s) | Retrieved by GA-IPPF (g/s) |
|---|---|---|---|---|
| Agrar Hauser | 312_01 | $CH_4$ | 0.31±0.03 | 0.3±0.03 |
| EPA | STR_6061411_01 | $CH_4$ | 0.60 | 0.57±0.04 |
| Prairie Grass | 57 | $SO_2$ | 101.5 | 104.7±3.7 |

Table 4 shows the emission rates and uncertainties through GA-IPPF in control release experiments, and the reported emission rates. The average difference between retrieved emission rates and reported ones is 3.8 %, which indicates the high accuracy of GA-IPPF in quantification estimation.

[Figure]

**Fig.10.** Rebuild the gases diffusion based on retrieved parameters in control release experiments; a1 and a2 are comparisons between simulated diffusion and actual samples in Agrar Hauser; b1 and b2 are comparisons between simulated diffusion and actual samples in EPA control release; c1 and c2 are comparisons between simulated diffusion and actual samples in Prairie Grass experiment.

  As is shown in Fig.10, the rebuild diffusions of gases by GA-IPPF in three kinds of control release experiment are logical, consistencies of simulated gases concentration and actual samples are good(see Fig.10.a1, Fig.10.b1 and Fig.10.c1,), each peaks of the samples in control release experiment could be reconstructed. The correlations between simulated gases concentration and actual samples are larger than 0.65, and the RMSE are within 2.7% (relative to the mean value of the selected samples' concentration). In general, reconstruction of gases concentration in both mobile-platform and UAV-based data are worse than that in in-situ network. Collected data in in-situ network are usually the mean value of a certain time, like 10 min in Prairie Grass emission experiment, which provide stabile inputs data in GA-IPPF, especially concentration samples. While the concentration samples in mobile-platform and UAV-based Aircore experiments is instantaneous, which may be inaccurate and exist fluctuations in collections. The advantages of vehicle-based and UAV-based sample systems are flexibility, which could acquire the

distribution of gases around the target monitoring sources freely. In-situ network is complicated to implement and the cost is high, and the wind direction should be considered when deploying. But its high stability and accuracy could helps us to quantify emission source. Therefore, environmental protection departments can choose detection systems according to actual emission monitoring needs. ''

2.The issue of quantification of methane from the energy sector has the potential to become potentially political and legal – with the concept of emissions levies or preferred contracts based on quantified methane emissions being used as a legal instrument. It is therefore of upmost importance that the authors of novel methodologies for quantification such as those shown here are aware of how their methods may be utilised in the future, potentially by 3rd party commercial companies, and ensure that any issues are fully declared (e.g. why are only 2 flights used to verify the method, were there occasions when this method failed, and why? There are 15 flights stated on the experiments section). They should also check that all references to accuracy are truly discussing accuracy (closeness to a true value) – and not precision (closeness to another measurement or model).

Thank you for your useful suggestion, it's critical to note that the requirement of the collected data by different measured instruments in GA-IPPF. Because the national government may develop different sample systems based on different platforms to monitor the methane emission. If GA-IPPF become a criterion in government monitoring policy, the limitation and application of GA-IPPF should also be introduced in this work to validate the reliable.

  In the latest revised version, we expounded the matters needing attention and process of GA-IPPF for users, which as follows:

1. Meteorological instruments should be equipped during the concentration sampling, which could acquire wind speed, wind direction, humidity and atmospheric pressure.

2. The wind speed should not be too low, and we recommended it should be larger than 2m/s.

3. During actual experiments, after the stable wind speed and wind direction were measured, the UAV-based Aircore system should try to fly along the cross section perpendicular to the wind direction.

Actually, we totally collected 15 Flights data around Pniówek coal on 18/08/2017 and 21/08/2017. Therefore, we initially considered choosing one Flight to discuss mine methane emissions detailly using the GA-IPPF model as an example.

  In this work, the Flights meet the two criterions has been selected in the analysis of methane quantification. Firstly, the wind speed is larger than 2 m/s, which would possible lead stable wind direction. Secondly, the flight trajectory of UAV is nearly perpendicular to the wind direction (within 15°). Finally, eight Flights fulfilled the set requirements simultaneously.

   As you mentioned that we should compare the emission rate calculated by GA-IPPF with actual known Pniówek coal emission rate in same time. It's really reasonable, however, we didn't know actual instantaneously methane emission of Pniówek coal during the Aircore Flight. Therefore, we prove the accuracy of GA-IPPF through indirect comparison, mainly including two parts:

   1. Comparison with other quantification methods, including emission-factories, NLSL, IPPF as well as mass balance and Gaussian inversion. The results calculated by different methods show differences, and the results calculated by GA-IPPF and these evaluation methods are in the same

order of magnitude among the retrieved emission rate, and between the maximum and minimum values of retrieved emission rates.

2. We evaluated the accuracy of GA-IPPF in three release-controlled experiments, which could evaluate the performance of GA-IPPF when actual emission rates are known, and the bias between retrieved emission rates and reported ones within 5.0 %, see table 4.

**Table 4 Performance of GA-IPPF model in different control release experiments**

| Database | Number | Gas | Release rates (g/s) | Retrieved by GA-IPPF (g/s) |
|---|---|---|---|---|
| Agrar Hauser | 312_01 | $CH_4$ | 0.31±0.03 | 0.3±0.03 |
| EPA | STR_6061411_01 | $CH_4$ | 0.60 | 0.57±0.04 |
| Prairie Grass | 57 | $SO_2$ | 101.5 | 104.7±3.7 |

3.If this method is suitable, I think a comment section needs to be added discussing other sorts of data that the team would expect this method to work with. I'm assuming that on-board drone measurements downwind from landfill / industrial sites would be a good option, would mobile measurements from vehicles potentially work?

[Figure]

Thank you for this suggestion. According to the characteristic of GA-IPPF model, it can be seen that the input data mainly includes location information of the samples, the concentration of the samples and the meteorological parameters. As long as the researchers or the energy monitoring department can obtain these collection data, GA-IPPF can be applied to quantify gases emission of industrial sites, gas leak from gas tank, etc.

[Figure]

**Fig.9.** Application of GA-IPPF in quantifying emission source of gases through different sample systems;including UAV-based Aircore system, ground-based In-situ network and mobile collection system.

In order to better demonstrate the applicability of this method, we have added a chapter named " Application of GA-IPPF", showing the performance of GA-IPPF in three different release-controled experiments, as follows:

As for the real experiments, we added three release-controled experiments as verification. The three experiments are UAV-based, vehicle-mounted and in-situ net respectively. The emission sources are spatially sampled, and then the emission intensity is retrieved by the quantitative GA-IPPF. Among them, the emission deviation between the intensity obtained by GA-IPPF and the actual release value are 3.2 %, 5.0 % and 3.2%, respectively, which proved a good quantitative effect.

Detailed responses in our manuscript are as follows:

"**Emission Estimates in control release experiment**

To evaluate the performance of GA-IPPF in controlled release experiments, we quantified the emission rates in release experiment through different gases sample systems, including UAV-based Aircore system, mobile sampling system and ground-based in-situ network. Detailed introduction of the concerned release experiment are as follows:

**Agrar Hauser control release**

This $CH_4$ release experiment was conducted on Agrar Hauser field near Dübendorf, Switzerland. The controlled methane source was release from a 50 L high-pressure cylinder, the height of this artificial source is 1.5 m. meteorological information were acquired by 3D anemometers around the emission source. UAV-based sample systems used in these release experiments contains two sensors, including Quantum cascade laser spectrometer (QCLAS) and active Aircore. It carried a series active measurements from 23 February to 14 March 2020.There is no other $CH_4$ source around Agrar Hauser field and the topography is flat. In this section, active Aircore $CH_4$ samples on 12 march 2020 (312_01) are chosen to use GA-IPPF to quantify methane release rate.

**EPA methane control release**

Environmental Protection Agency (EPA),USA developed OTM 33A method to quantify oil and gas leakage based on mobile measurement platforms, which consist $CH_4$ in-situ sensor (G1301-fc cavity ring-down spectrometer (Picarro)), a collocated compact weather station and a Hemisphere

Crescent R100 Series GPS system. The accuracy of in-situ sample is within ±5%, and in-situ sensor was implemented at height of 2.7 m based on vehicle. Weather station provides atmospheric temperature, pressure and humidity, as well as 3-D wind direction and wind speed. Simulated $CH_4$ leakage source was conducted using 99.9% methane high pressure cylinders as the gas supply. Total 20 experiments of control releases were published by EPA to evaluated OTM 33A.

**Prairie Grass emission experiment**

Prairie Grass emission experiment was mainly to evaluate the diffusion of $SO_2$ from point source under different meteorological circumstances [Barad, 1958], the height of emission source is 0.46 m, all in-situ sensors are set at heights of 1.5 m. $SO_2$ concentration sampled by in-situ network at radius of 50 m,100 m,200 m, 400 m and 800 m. Samples in R57 release (10-minute sampling periods), total 94, were selected to quantified $SO_2$ emission rate from release instrument. Reported emission rate of $SO_2$ in R57 is 105.1 g/s, samples collected in radius of 800 m was neglected in this discussion for amount of what is extreme less. Reported wind speed is 4.85±1 m/s, wind direction is 184±10°.

**Table 4 Performance of GA-IPPF model in different control release experiments**

| Database | Number | Gas | Release rates | Retrieved by GA-IPPF |
|---|---|---|---|---|
| Agrar Hauser | 312_01 | $CH_4$ | 0.31±0.03 g/s | 0.3 |
| EPA | STR_6061411_01 | $CH_4$ | 0.60 g/s | 0.57 |
| Prairie Grass | 57 | $SO_2$ | 101.5 g/s | 104.7 |

Table 4 shows the emission rates and uncertainties through GA-IPPF in control release experiments, and the reported emission rates. The average difference between retrieved emission rates and reported ones is 3.8 %, which indicates the high accuracy of GA-IPPF in quantification estimation.

[Figure]

**Fig.10.** Rebuild the gases diffusion based on retrieved parameters in control release experiments; a1 and a2 are comparisons between simulated diffusion and actual samples in Agrar Hauser; b1 and b2 are comparisons between simulated diffusion and actual samples in EPA control release; c1 and c2 are comparisons between simulated diffusion and actual samples in Prairie Grass experiment.

As is shown in Fig.10, the rebuild diffusions of gases by GA-IPPF in three kinds of control release experiment are logical, consistencies of simulated gases concentration and actual samples are good(see Fig.10.a1, Fig.10.b1 and Fig.10.c1,), each peaks of the samples in control release experiment could be reconstructed. The correlations between simulated gases concentration and actual samples are larger than 0.65, and the RMSE are within 2.7% (relative to the mean value of the selected samples' concentration). In general, reconstruction of gases concentration in both mobile-platform and UAV-based data are worse than that in in-situ network. Collected data in in-situ network are usually the mean value of a certain time, like 10 min in Prairie Grass emission experiment, which provide stabile inputs data in GA-IPPF, especially concentration samples. While the concentration samples in mobile-platform and UAV-based Aircore experiments is instantaneous, which may be inaccurate and exist fluctuations in collections. The advantages of vehicle-based and UAV-based sample systems are flexibility, which could acquire the

distribution of gases around the target monitoring sources freely. In-situ network is complicated to implement and the cost is high, and the wind direction should be considered when deploying. But its high stability and accuracy could helps us to quantify emission source. Therefore, environmental protection departments can choose detection systems according to actual emission monitoring needs.

In summary, GA-IPPF could apply in different sample systems to quantify emission rate from strong sources with high flexibility.

**Minor**

I am far from a professional proof-reader, and would recommend that this manuscript is looked over by a proofer after corrections as there are numerous tense and grammar issues that need resolving to make the paper read as desired.

Dear reviewer, we are sorry for our wrong tense and grammar in our original manuscript. And we have checked and modified these wrong parts in our latest manuscript, and we have inquired Copernicus Publications for the recommended Professional English editing company. They informed once our manuscript be accepted for final publication, they will perform an extensive copy-editing for English with the goal to ensure the paper adheres to scientific writing standards, grammatical accuracy, and overall readability.

L20: There are plenty of monitoring methods – but very little verified quantification methods suitable for coal mines.

Thank you for your suggestion on this sentence, we have modified this sentence to make the statement more reasonable as follows:

"There are plenty of monitoring methods to quantify gases emission rate based on gases concentration samples around the strong sources. However, there is a lack of quantitative models to evaluate methane emission rate from coal mines with less priori information."

L34: Grammar. Release is concerning.

Thank you for your kindly comment on this sentence, we have modified this sentence as follows:

"The release of $CH_4$ into the atmosphere during coal mining is very concerning because it contributes to increased atmospheric concentration of $CH_4$, one of the most important greenhouse gases and is a waste of resources."

L38: Why are BU only useful for strong sources?

We apologize for this wrong expression, actually, besides strong emission sources, BU can provide also the emission rate or gases flux of grids with different spatial resolution. And we has modified this sentence as follows:

"Bottom-up inventories can provide us with $CH_4$ emission rates from strong point sources or gridded $CH_4$ fluxes with different spatial resolutions, which play a great role in statistical analysis."

L42: This seems to be a common misuse / expectation of a BU inventory. They are not intended to be able to capture variable emissions, but are a statistical average expectation. It is only worth considering inventories compared to spot measurements if there is some valid statistical analysis (either temporal, or numbers of sites)

We're sorry for this inappropriate expression. BU does play an important role in statistical analysis, such as comparison of different national emission rates, statistical basis for environmental protection policies and energy distribution in different industry sectors, et. al. We wrongly expressed negative appraisal on its function in real-time emissions evaluation.

Therefore, we have modified the original text as follows:

"Bottom-up inventories can provide us with $CH_4$ emission rates from strong point sources or gridded $CH_4$ fluxes with different spatial resolutions, which play a great role in statistical analysis. However, the low temporal resolution of inventory data does not allow us to obtain emission intensity from target sources instantaneously."

L44: What improvements have suddenly made this possible?

Sorry for this incomplete expression, we have modified this sentence as follows:

"With the development of different atmospheric $CH_4$ concentration measurement techniques, like Fourier spectrometer, differential absorption Lidar, Aircore system, and in-situ sensors, $CH_4$ emission rates for strong emission sources could be quickly quantified by top-down methods with high accuracy."

L46: Tense: should be is capable of obtaining.

Thank you, and we have revised this sentence as follows:

"Greenhouse gases observing satellite (GOSAT) and TROPOspheric Monitoring Instrument (TROPOMI) are capable of obtaining the column concentration of $CH_4$ ($XCH_4$, ppb) with spatial resolution of 10 km×10 km and 5 km×7.5 km respectively."

L59: But most aircraft equipped with any $CH_4$ sensor are able to achieve ppb precision and are perfectly capable of measuring downwind flight plans from coal mines. Many aircraft are capable of this such as Scientific Aviation et.

We are sorry for this wrong comment on the aircraft remote sensing systems in our original manuscript. It's correct that most aircraft equipped with $CH_4$ sensors are able to achieve ppb precision and are perfectly capable of measuring downwind $CH_4$ of coal mines. Such as Methane Airborne MAPper (MAMAP) instrument(Krautwurst et al., 2021), and airborne-based AVIRIS-NG system, both of them could acquire the concentration in downwind direction with reasonable flight track. We have modified sentence in Line 59 as follows:

"An airborne vehicle could fly at low altitudes to improve the acquisition of $CH_4$ concentration (Elder et al., 2020; Wolff et al., 2021; Krautwurst et al., 2021) and estimate $CH_4$ emission from strong sources by the cross-sectional flux method or the Gaussian dispersion method. However, the cost of airborne experiment is high and the fly plan is easily to be restricted by aviation control policies."

L62: What about ground based eddy covariance?

We are sorry for neglecting the introduction for ground-based eddy covariance, and we have added this information in our latest manuscript as follows:

"Ground-based eddy covariance sites could monitor agriculture and forestry ecology methane flux with high temporal resolution(Jha et al., 2014), such as mangrove ecosystem, larch forest in eastern Siberia(Nakai et al., 2020)."

General: to L70. This section feels unnecessarily negative about the capabilities of the rest of the scientific community with regards to being able to measure emissions from coal mines. There are several methods discussed here that I would envisage perfectly capable of making precise measurements that could enable a quantification of emission estimate.

Sorry for the unnecessarily negative comments we made on different detection technologies in the original manuscript. In the latest submission, we have introduced the application, advantages and disadvantages of each detection technology in quantifying methane emissions. As we know, any detection technique has its own unique advantages and limitations. We have re-edited this paragraph in latest submission as follows:

"Greenhouse gases observing satellite (GOSAT) and TROPOspheric Monitoring Instrument (TROPOMI) are capable of obtaining the column concentration of $CH_4$ ($XCH_4$, ppb) with spatial resolution of 10 km×10 km and 5 km×7.5 km respectively. The regional $CH_4$ flux can be retrieved by assimilating the measured $XCH_4$ into an atmospheric dispersion model (Tu et al., 2022; Feng et al., 2016). PRISMA hyperspectral imaging satellite and GHGsat can detect increased $CH_4$ caused by strong emission sources with high spatial resolutions, and the comprehensive $CH_4$ emission can be quantified by integrated mass enhancement or cross-sectional flux method (Guanter et al., 2021; Varon et al., 2020). It plays a huge role in the analyzing methane emission rate from strong sources, but it has high requirements for satellites' detection track, that is, to monitor the methane distribution in the target area within coverage range (Schneising et al., 2020; Varon et al., 2019). Airborne sensors can fly at low altitudes to improve the acquisition of $CH_4$ concentration data and estimate $CH_4$ emission from strong sources by the cross-sectional flux method or the Gaussian dispersion method (Elder et al., 2020; Wolff et al., 2021; Krautwurst et al., 2021). It enables repeated monitoring of emission sources in a large area in a short period of time, however, airborne experiments' cost is high and the flight tracks may be restricted by the aviation control policies. Ground-based eddy covariance sites can monitor agriculture and forestry ecology methane flux with high temporal resolution, such as mangrove ecosystem(Jha et al., 2014), larch forest in eastern Siberia(Nakai et al., 2020). Its accuracy is very high, but there is currently less monitoring of methane emissions from strong point sources using eddy covariance. When ground-based concentration sensors fixed in appropriate position, they have the advantage of continuously sampling gas concentration in downwind direction from the source. It will provide important dispersion data for methane emission quantification model at the enterprise level, but these sensors usually need to be carried on a vehicle platform to obtain methane concentration at different locations (Zhou et al., 2021; Robertson et al., 2017; Caulton et al., 2017). Ground-based differential absorption LIDAR can obtain the $CH_4$ profile concentration in different altitudes, whose data is suitable as the input of the emission-retrieval model (Shi et al., 2020a), but it has high requirements in terms of hardware performance and system stability (Shi et al., 2020b). An unmanned aerial vehicle (UAV) can reach any location rapidly around the $CH_4$ sources, which can sample $CH_4$ concentration with location information (Nathan et al., 2015; Iwaszenko et al., 2021), when equipped with concentration sensors. It can acquire the distribution characteristics when sufficient concentration data are collected, which is beneficial to retrieving emission rate. The cost of UAV-based AirCore system is low and the process of its sample data is relatively simple, but the diffusion of methane emitted from strong sources may be sampled incompletely."

L73: What does high applicability mean in this sense?

Sorry for the unclear expressing here. Actually, what we want to express is that the most outstanding advantage of Aircore system is collecting methane concentration at different locations freely, thus it can obtain useful data to rebuild emission diffusion, contains spatial location and corresponding concentration. Then, GA-IPPF have a self-adjusting effect on the influence of errors in detection accuracy and the acquisition of meteorological conditions. See the figure below for details, all of which have the effect of reducing errors.

[Figure]

**Fig. 8.** Influence of accuracy of parameters on retrieved emission results. The baseline represents the emission rate setting of $CH_4$, 300 g/s: (a) wind speed, with additional error ranging within 0.2–2 m/s and an interval of 0.1 m/s, (b) wind direction, with additional error ranging within 5°–50° and an interval of 5 °, (c) accuracy of $CH_4$ samples, with additional error ranging within 0.5%–5.0% and an interval of 0.5%, and (d) amount of $CH_4$ samples, randomly selected as 20–90 among the defined 99 samples.

We have revised this sentence as follows:

"In 2017, we developed an UAV-based active AirCore system, which could sample spatial atmospheric $CO_2$, $CH_4$, and CO with high accuracy (Andersen et al., 2018), aiming to retrieve greenhouse gases emission of strong sources. The most urgent issue we need to address is to develop an emission quantification model to make use of the advantage of the collected data by Aircore, namely collecting data at different locations with high flexibility."

L74: How will it have less uncertainty if the inputs still have the same uncertainties attached to the actual measurements? The discussion to L83 makes it seem that these important atmospheric parameters can be discarded in favour of a set of model parameters? From experience at controlled release experiments small changes in wind direction and other meteorological conditions can have a dramatic effect on the plume behaviour.

Dear reviewer, this question you raised is very meaningful. According to the formulations of the Gaussian Diffusion Model, small changes in wind direction and speed, as well as other meteorological conditions, can indeed cause significant impact on the final retrieved emission results.

The original meaning of sentence (line L83), which also highlights the shortcoming of the existed methods, once there are errors in meteorological data, it will cause great impact on the final assessment. In Fig 8, we discussed the influence of different uncertainty of GA-IPPF in wind speed and wind direction on the final calculation through simulations. It worth nothing that during the process of optimization, the uncertainty of the original input of wind speed and wind direction has been added. For example, wind speed was fluctuated within ±2m/s and wind direction was fluctuated within ±60°. During the process of GA-IPPF, upper and lower limits are set for wind

speed and wind direction. For example, if the original wind speed is 3m/s, original wind direction is 80°, the lower limit of wind speed is 1m/s, and the upper limit is 5m /s, the lower limit of the wind direction is 20°, and the upper limit is 140°. The wind speed will have a linear relationship with the final diffusion; the wind direction determines the location of the sampling point, and the concentration of sampling points in different locations represent the characteristic of diffusion from emission source. Because GA-IPPF will calculate the global optimal solution for all the unknown parameters, so important parameters such as wind speed and wind direction will be dynamically adjusted within lower and upper limits. The objective function we set is served as the judgment of GA-IPPF, that is, the reconstructed data is consistent with the real sampled data. Through the optimal adjustment of GA-IPPF, the objective function would meet its minimum value. In actual measurement, these meteorological parameters cannot be discarded in theory. However, in the case of insufficient funds or technology, some researchers or energy departments have not installed meteorological instruments in field experiments. In this situation, the alternatives are given in section 3.3, and we recommend meteorological reanalysis data to participate in GA-IPPF. Through comparison, it is found that the deviation between the emission intensity obtained by using meteorological data and meteorological analysis data is within 20%. We believe that once the limits of each parameter are set appropriately, the accuracy of inversion can achieve satisfactory results. The advantage of the GA-IPPF model is that all unknown parameters in Gaussian dispersion model are dynamically adjusted, reducing the accuracy requirements of the collected data in actual experiments.

L106: Is the accuracy only 20ppb using a G2401-m? This is concerning, why is it so large? What is the location accuracy of the sampling, how much does the sample bleed into itself over the course of a flight? Is the mixing in the Aircore linear across flight time?

Indeed, the precision of methane measurements was 20 ppb, because we used a high-range $CH_4$ mode of the CRDS, and the normal mode would not be able to measure up to 125 ppm. This is actually not an issue as the relative uncertainty of 20 ppb with respect to 100 ppm is only 0.02%. The spatial resolution of the AirCore measurements is estimated to be 18-40 m with a UAV flight speed of 1-2 m/s. Since the spatial resolution is dominated by the sample smearing in the cavity of the CRDS analyzer and the air samples were analyzed quickly after landing, the spatial resolution was only affected by the UAV flight speed.

L117: correlation not connection? How good a correlation?

Dear reviewer, thank you for this suggestion, we are sorry for we haven't expressed this sentence clearly. We have modified this sentence in latest manuscript as follows:

"During the detection, the spatial position of the radiosonde is well recorded by satellite, we assume that the uncertainty in the wind direction is low."

L135: I'm not 100% sure what alpha represents, please clarify (with a reference if possible)

[Figure]

Fig R3. Introduction to α in Gaussian dispersion model

Dear reviewer, α is the reflected index of the diffusion gases when the gas molecular touch the ground, it can be simply treated as the ejection effect. For example, if the gas molecular catapult into the atmosphere totally, α =1. Conversely, if the gas molecular absorb by ground totally, no ejection effect, α =0. α is relative with the actual field circumstance and the diffusion molecular.

Principle:

In Fig.R3, we take $CH_4$ dispersion as an example. H is the effectively emission height of real source, assuming an imaginary source symmetrical to the ground. The imaginary is also regarded as a potential emission source, the sensor sample concentration of $CH_4$ is the sum of a and α×b. therefore, α would determine the value of samples in certain position. You could reference this item according to page 36 in https://ansn.iaea.org/Common/Topics/OpenTopic.aspx?ID=13012.

Approx. L160. Is there prioritization in the fitting process? Are some variables given priority due to their certainty? General question about the process, if the inputs are very close to the outputs, then I presume that the standard gaussian plume quantification would also be very close?

Dear reviewer, thank you for this comment. The concept similar to priority is existed in our proposed model, refers to the upper and lower bounds we set for each unknown parameter. For example, during the actual experiments, if uncertainty of wind speed is relatively small, we would set smaller range for its dynamic adjustment, which means we set larger weight constraints for wind speed, which is similar to the concept of priority of wind speed in final emission quantification.

The relationship between value of inputs and output value cannot directly judge whether the Gaussian quantification is reliable. In GA-IPPF, input parameters in final step of emission calculating process are obtained according to the repeated fitting results by genetic algorithm. During the solution of IPPF, we set fluctuation for each parameter based on the calculated results by genetic algorithm. The final output values are retrieved according to the judgment condition we set, refers to the relationship between the rebuild diffusion calculated by retrieved parameters and actual samples. Therefore, we don't judge the accuracy of Gaussian plume quantification based on the consistency of the input and output values. In fact, RMSE and $R^2$ between the reconstructed

diffusion concentration and the actual samples are worked as judgement, when the RMSE is small and R2 is high, that means, the Gaussian quantification for the emission sources is better.

L190: If these are spiral patterns, then the figure doesn't make this very clear. Can these be replotted to make it clear if that is the case?

The samples of Aircore is the concentration of methane in certain position, it is not a continuous data acquisition. The sample trajectory of UAV is nearly a cross-section on two-dimensional plane, which cannot show the samples with obvious 3d Visual. In latest submission, we tried our best to change the visual angle to make Fig 3 closer to 3D visual as follows:

[Figure]

Para 218: This paragraph needs tidying up, it is unclear as it currently reads and has numerous typos. The reconstruction is impressive – especially with two peaks in Flight 15 so it is important that this section is as simple and clear as possible.

Thank you for this suggestion, we have modified this paragraph in latest submission. And as you mentioned the two peaks in Figure 3 (b), it was accused by the diffusion characteristic from strong point sources, in order to make it easier to understand, we simulated the diffusion at different altitudes as follows:

Table R1. Parameters settings

| Parameters | value |
|---|---|
| Wind speed (m/s) | 2.5 |
| Wind direction (°) | 90 |
| Emission rate (kg/s) | 300 |
| a | 0.11 |
| b | 0.93 |
| c | 0.1 |
| d | 0.93 |
| H (m) | 20 |
| B (ppb) | 2000 |

We also carried simulation experiments to show vertical distribution at different heights. As is shown in Fig.R2, it would exist different hot spot in different height.

In Fig.R2, we could see diffusion of gas exist higher value in each height, for example, the diffusion of 50m and 150m are in same magnitude. Fig 3b is show the diffusion of $CH_4$ in height of

22.3 m and 52.4 m, the two spot are around the wind direction (X axis), thus, it exist two spots in Fig 3 b. If the UAV's Flight across this two heights, it would be possible that the samples exist two hots, which would be explained for the two peaks in Fig. 3(b).

[Figure]

Fig.R2 simulated diffusions of $CH_4$ at different heights, a.50 m; b.100 m; c.150 m and d. 200 m.

L250: If the Gaussian plume doesn't account for background, has the dataset been adjusted to make that correction?

Sorry for the incorrect expression this sentence in our origin manuscript, what we actually want to say is that other researchers did not consider the background value as an unknown parameter in their emission quantification model. They usually extract the background value through some statistical analysis methods, for examples, Nassar *et al* has used the OCO-2 data to evaluate $CO_2$ emission from power plant successfully, the principle of quantify model is Gaussian dispersion, he regarded the value of background concentration of $CO_2$ referenced by four plausible manually-selected background regions(Nassar et al., 2021).   In Andersen's study, the background concentration of mehane was regarded as the sampled concentration that not effected by the plume in each Flight. And we have modified this sentence as follows:

  "The background concentration of mehane was regarded as the sampled concentration that not effected by the plume in each Flight."

Nassar, R., Mastrogiacomo, J. P., Bateman-Hemphill, W., McCracken, C., MacDonald, C. G., Hill, T., O'Dell, C. W., Kiel, M., and Crisp, D.: Advances in quantifying power plant CO2 emissions with OCO-2, Remote Sens. Environ., 264, 10.1016/j.rse.2021.112579, 2021.

Approx L265: I don't know if this is possible, but it would be incredibly helpful if there was some sort of visualization of the $CH_4$ atmospheric concentration for each of the methods of quantification (e.g. what does the plume look like in 2-D so that the

variability can be understood). This would be most helpful for where there is significant differences between the methods to show what the plume visualisation looks like in each of the quantifications.

Thank you for this suggestion, we have added this part in our supplement, which is shown as follows:

In this section, we rebuild the 2-D plume of $CH_4$ dispersion in Flight 6 and Flight 15 based on the different methods, including NLSF, GA-IPPF, inventory, Mass balance and Gaussian inversion. It worth nothing that 2-D plume rebuild by NLSF, GA-IPPF and inventory are according to formula 1. Relative diffusion parameters in inventory 2-D rebuild plume are same as that retrieved by GA-IPPF in each Flight. 2-D plume rebuilt by Masa balance are according to formula 10. 2-D plume rebuilt by Gaussian inversion is according to formula 13.

[Figure]

L330 (approx.) General discussion of measurement stability – the variation in wind parameters, disturbance in the airflow from the drone, changes in plume behaviour are all potential problems for quantification. Are the random errors used here sufficient to capture the potential uncertainty and can they be justified with reference to uncertainty from other studies?

Thanks for this suggestion, variation in wind parameters, disturbance in the airflow from the drone, changes in plume behavior, these items exist in the actual measurements. In the OSSE, firstly, for the variation of wind parameters, we regarded that the random error can represent the uncertainty in the actual emission quantitative evaluation, because we added random errors to the wind speed and wind direction in the simulation experiment, which are randomly distributed within a certain range. For example, if the uncertainty of wind speed is 0.5 m/s, the real value of wind speed is 2.5m/s, wind speed during the measurement may corresponds to 2.2 m/s, 2.8 m/s or 3 m/s. This would represent characteristic of wind speed in the actual measurement. And for the UAV airflow disturbance and plume behavior you mentioned, we thought that they could cause change of the sampling concentration in different locations, thus, we regard the added random error in concentration collection is the combinate effect of sample error of Aircore system itself, UAV airflow disturbance and plume change. Therefore, we believe that the random error incorporated in the simulation experiment can be used to assess the uncertainty in the emission quantification.

However, both the solution process in GA and IPPF are not linear, we couldn't define the uncertainty through formula based on uncertainties of the input data.Based on your suggestion, we have also modified the uncertainty calculation method as well as the principle, this suggestion really help us to make our uncertainty evaluation more reasonable, which is shown as follows:

"**Uncertainty Analyses**

The GA-IPPF model will be calculated 1000 times repeatedly based on the collected samples of $CH_4$ concentration, then, the uncertainty and final retrieved emission rate could be defined by

$$\sigma = \sqrt{\frac{\sum_{i=1}^{N}(q_i - \bar{q})^2}{N}} \tag{8}$$

$$q_r = \bar{q} = \frac{1}{N}\sum_{i=1}^{N}q_i \tag{9}$$

σ is the uncertainty of retrieved emission rate;$q_i$ is the i *th* retrieved emission rate, i=1,2,3…1000; $\bar{q}$

is the mean value of the $q_i$; N is 1000; $q_r$ is regarded as the value of retrieved emission rate. The values of other parameters (a,b,c,d,H,Ws,Wd,B,α) calculated by GA-IPPF are also defined in same principle."
L381. The claim that this result would guarantee emission calculation accuracy to better than 99.2% seems very over confident. This type of statement brings me back to the knowledge that methane emissions may be used as legal instruments in the near future and claims of such accuracy when not compared to a blind control is not acceptable. There needs to be a clear rethink of the use of accuracy, error and precision throughout so that it is clear what is being compared. As there is no "known" value, no claim on accuracy can be made.

We are sorry for this simple definition. In origin submission, we declared that the accuracy of GA-IPPF better than 99.2% is based on simulated experiments, which is unreasonable. Therefore, we have selected several release-controlled experiments to evaluate the uncertainty of emission rate calculated by GA-IPPF model, based on the publicly available concentration sampling datasets, as well as meteorological parameters, including wind speed, wind direction and humidity. These release-controlled experiments are similar to the Aircore experiment in Pniówek coal mine, which

are reliable to test the performance of GA-IPPF.

In addition, we also updated the uncertainty analysis method to fully consider the effects of various variables on the final results during the experiments. The specific assessment results are shown as follows:

"**Uncertainty Analyses**

The GA-IPPF model will be calculated 1000 times repeatedly based on the collected samples of $CH_4$ concentration, then, the uncertainty and final retrieved emission rate could be defined by

$$\sigma = \sqrt{\frac{\sum_{i=1}^{N}(q_i - \bar{q})^2}{N}} \tag{8}$$

$$q_r = \bar{q} = \frac{1}{N}\sum_{i=1}^{N} q_i \tag{9}$$

$\sigma$ is the uncertainty of retrieved emission rate; $q_i$ is the i *th* retrieved emission rate, i=1,2,3…1000; $\bar{q}$

is the mean value of the $q_i$; N is 1000; $q_r$ is regarded as the value of retrieved emission rate. The values of other parameters (a,b,c,d,H,Ws,Wd,B,$\alpha$) calculated by GA-IPPF are also defined in same principle."

Thus, we found that the bias of quantified emission rates calculated by GA-IPPF in release-controlled experiment is larger than that in the simulated experiment.  In order to express GA-IPPF's performance rigorously in actual experiment, we presented the uncertainty of quantification results of coal mine Pniówek coal mine shaft in latest submission.

Based on the new result in uncertainty analysis, we also modified this sentence in latest manuscript as follows:

"The UAV-based AirCore system can acquire more than 99 $CH_4$ samples in actual feasible measurements, therefore, it is believed that accuracy of $CH_4$ samples (>95.0 %) collected by the AirCore system bring less influence in theory."

L401. The conclusions feel very generic and non-specific in large parts to the work shown here, they should be reconsidered with the value added of GA-IPPF in mind, rather than generalities of coal emissions. It is good to see that the model is being considered for other uses too, is there any possibility that this could be expanded on in the main text to demonstrate how this would be done for vehicles?

Thank you for your comments on the section of Conclusion, as this manuscript is submitted to special issue of Comet, in original version, we highlighted GA-IPPF and the UAV Aircore system for quantitative assessment of methane emission from coal mining.

In our latest submission version, we presented the framework of GA-IPPF in quantifying gases emission sources, such as factories, gas tank,etc.

[Figure]

**Fig.9.** Application of GA-IPPF in quantifying emission source of gases through different sample systems;including UAV-based Aircore system, ground-based In-situ network and mobile collection system.

This figure shows target emission sources detectable by GA-IPPF through concentration sampling system based on different platforms, which helps readers to understand the potential usefulness of GA-IPPF.

We also demonstrated that GA-IPPF can perform a quantitative assessment of emissions based on sampling data from UAV-based Aircore system, vehicle-mounted, and ground-based in-situ networks through publicly controlled release experimental datasets. See Table 4.

**Table 4 Performance of GA-IPPF model in different control release experiments**

| Database | Number | Gas | Release rates (g/s) | Retrieved by GA-IPPF (g/s) |
|---|---|---|---|---|
| Agrar Hauser | 312_01 | $CH_4$ | 0.31±0.03 | 0.3±0.03 |
| EPA | STR_6061411_01 | $CH_4$ | 0.60 | 0.57±0.04 |
| Prairie Grass | 57 | $SO_2$ | 101.5 | 104.7±3.7 |

Table 4 shows the emission rates and uncertainties through GA-IPPF in control release experiments, and the reported emission rates. The average difference between retrieved emission rates and reported ones is 3.8 %, which indicates the high accuracy of GA-IPPF in quantification estimation.

Therefore, in the latest submission, we have revised the framework of the conclusions and stated the performance of GA-IPPF in different sampling systems as you proposed. The potential application of GA-IPPF is also highlighted in latest Conclusion. Details are shown as follows.

"In this study, we present a quantified model for strong point emission source based on concentration sampling data, named GA-IPPF. During CoMet campaign in 2017, we successfully monitor methane emissions from a ventilation shaft in Pniówek coal mine through the concentration data measured by UAV-based AirCore system. Results show that $CH_4$ emissions rate from ventilation shaft are not consistent even in a short time.

GA-IPPF can reconstruct the concentration dispersion around the point emission source, and the largest $R^2$ between the measured $CH_4$ concentration and the reconstructed concentration in the selected 8 Flights around Pniówek coal mine can reach 0.99.

In Observing System Simulation Experiments (OSSE), we discuss the sensitivity analysis of different parameters setting on final retrieved emission rate by GA-IPPF. We demonstrate that GA-IPPF has self-adjust function to achieve an optimal solution on emission rate, which will reduce the requirements for hardware performance in actual emission quantification experiment.

We also test the performance of GA-IPPF in three control release experiments with different sampling devices, including vehicle-mounted in-situ system, UAV-based AirCore system and ground-based in-situ network observation, and the biases between retrieved emission rates and reported ones within 5.0%.

In future, GA-IPPF has great potential in the point-source quantitation based on mobile concentration sampling system, which can help to renew and enrich the gases emission inventories on strong point sources. "

---

## Author Comment (AC2)

The manuscript presents an interesting study of $CH_4$ emissions from coal mines, but it is not well written in parts and many small sections need to be corrected / clarified (see attached PDF). For example, it is not clear what is meant by effective emission height and why it is different between the 2 flights when the emission point source (the stack) remains at the same height. There is very little discussion of the wider applicability of the technique. The abstract implies that it can be used to calculate emissions from coal mines, but the results section seems to suggest that it works only for single point sources, such as a vent or shaft, and that there would be big error bars on q if the emission was not from a point source. This needs to be clarified in the discussion. There should be some recommendation as to how many flights would be required to minimise the errors on the model results.

Dear Reviewer, thank you very much for all your comments on this manuscript, which are extremely helpful to improve the expression of our manuscript. We have revised the full text of the right based on the comments you marked in our manuscript PDF. And we make response to each comment one by one. Words in blue color are your comments, and our responses are in black color words.

**Effective emission height**

The effective height is related with emission intensity of the source, the emitted speed of the gas, the meteorological conditions (wind, atmospheric temperature, humidity and pressure, et al.) and solar radiation intensity. We illustrate it by Fig.R1 and Holland formulas.

As is shown in Fig.R1, H (m) is the effective stack height which is the sum of h and $\Delta h$, h (m) is the stack height, $\Delta h$ (m) is the plume rise height.

$\Delta h$ is calculated by the Holland formulas given in Equations 1 and 2.

$$\Delta h = \frac{u_s \times D_s}{u_0}(1.5 + 2.7\frac{T_s - T_a}{T_s}) = \frac{1}{u_0}(1.5 u_s D_s + 9.79 \times 10^{-6} Q_H) \qquad 1$$

$$Q_H = 0.35 \times Pa \times Q_v \times \frac{T_s - T_a}{T_s} \qquad 2$$

where $u_s$ (m/s) represents the exit speed of the $CO_2$ emitted from a stack, $D_s$ (m) represents the diameter of a stack, $u_0$ (m/s) is the mean wind speed at the height of the stack, $T_s$ (K) is the temperature of the emission from the stack, $Q_H$ (kJ/s) is the heat emission efficiency, and $Q_v$ (m³/s) is the actual emission rate of $CO_2$ from the stack.

As you mentioned that the emission heights of 2 Flights are different, because the selected 2 Flights were collected around the Pniówek coal mine on two different days, (a) August 18, 2017 and (b) August 21, 2017 respectively. The wind speed, emission intensity of methane are both different, which would lead to different emission height.

**Applicability of GA-IPPF**

Thank you for pointing that this method is not applicable to the quantification of gas emissions from non-point sources.GA-IPPF is designed to quantify methane emissions from strong point sources from coal mines, refers to ventilation shaft in this manuscript, and in order to avoid misunderstanding for readers, we have emphasized the type of application of this method in the our latest manuscript. For example, "strong point emission source" and "ventilation shaft".

In order to highlight the practical applicability of GA-IPPF model, we demonstrated performance of GA-IPPFmodel in quantifying strong point source by different sample systems:

[Figure]

**Fig.9.** Application of GA-IPPF in quantifying emission source of gases through different sample systems;including UAV-based Aircore system, ground-based In-situ network and mobile collection system.

**"Emission Estimates in control release experiment**

To evaluate the performance of GA-IPPF in control release experiments, we quantified the gases emission rates in release experiment through different gases sample systems, including UAV-based AirCore system, mobile sampling system and ground-based in-situ network. Detailed introduction of the concerned release experiment are as follows:

**Agrar Hauser control release**

[revised manuscript text omitted]

There should be some recommendation as to how many flights would be required to minimize the errors on the model results.

In this study, we focus on methane emissions from ventilation shaft in coal mine, which is treated as strong emission point source. The volatility of its emissions is significant. We assume that the intensity of methane emission is constant during the Aircore sampling collection. However, if methane emission rate is always fixed and a constant value, then multiple Aircore Flight would certainly improve the accuracy of quantified emission rate. Actually, methane emission rate from coal mine is always different between two Flights periods, so it is unreasonable to use multiple Flights to reduce emission error. We understand that you want us to present the data requirements of the Aircore system for the actual users, and we summarized the performance of GA-IPPF based on a certain number of samples. We recommended that the amount of samples in a single UAV-based Aircore system (with accuracy better than 99 .5%) larger than 90, the accuracy of retrieved emission rate would be better than 99 %, when error in wind speed is $\pm0.3$ m/s and error in wind direction is $\pm30°$.

**Detailed suggestions**

1. In section of Introduction, expression of "a Pniówek coal-mine ventilation shafts" should be modified.

   Dear reviewer, thank you for this suggestion, "a Pniówek coal-mine ventilation shafts" has been revised as "$CH_4$-emission rates from a ventilation shaft in Pniówek coal (Silesia coal mining region mine, Poland)."

2. In section of Introduction, expression of "are not allowed to repeat the quantification of $CH_4$ emission from coals in the same day" should be modified, Poor English and not clear - you mean that the orbital pattern of the satellite means that they do not pass over the same area twice within the same day.

Sorry for our mistake for this expression, and your understand is really correctly, the original meaning is orbital pattern of the satellites have low possibility to pass over the same area twice in same day.

we have modified this sentence in our latest manuscript as follows:

"but orbital patterns of the satellites have low possibility to pass over the same area twice in a single day, which would not allow to multiple quantify $CH_4$ emission from coals in one day"

3. In section of Introduction, "Most ground-based sensors have the advantage to sample the concentration around the source continuously," I presume you mean fixed location. This is also dependent on wind direction, so the emission will not be sampled all of the time, unless monitoring is inside of the source.

Thank you for this helpful suggestion, we are sorry for this incorrectly sentence. We strongly agree with you that ground-based equipment, such as vehicle-based monitoring system, and in-situ networking systems, need to collect data in downwind direction from emission source to be able to quantify the emission sources based on GA-IPPF. As shown in Fig 9.

[Figure]

**Fig.9.** Measured gases concentration in downwind direction by Ground-based sensors, include in-situ network and vehicle-based sample system.

We have modified this sentence according your suggestion as follows :

"When ground-based sensors fixed in appropriate position, they have the advantage that sampling gas concentration in downwind direction form the source continuously"

4. In section of Introduction, "data" should be added after "concentration"; the "high accuracy" means what? AirCore only collects the sample so presumably GPS location X.Y.Z co-ordinates at high frequency.

We have changed "concentration" to "concentration data" throughout in our latest manuscript.
Sorry for the wrong expression in our original manuscript, "high accuracy" means the accuracy of $CH_4$ concentration samples. Aircore actually only collects samples, which would be analyzed by cavity ring down Spectrometer model G2401-m. The spectrometer could promise the accuracy of $CH_4$

concentration during the Flight. Hence, we modified this sentence as follows:

"In 2017, we developed an UAV-based active AirCore system, which could sample spatial atmospheric $CO_2$, $CH_4$, and CO with high accuracy (Andersen et al., 2018), aiming to retrieve greenhouse gases emission for strong sources."

5. In section 2.1, Need to be clear that the AirCore sample is removed from the drone and replayed through the Picarro on the ground.

Thank you for this suggestion, we have added this information in our latest manuscript as follows:;

"After obtaining air samples from the drones during field campaigns, $CO_2$, $CH_4$ and CO collected by AirCore system would be analyzed by ground-based G1301-fc cavity ring-down spectrometer (Picarro)."

This item is really helpful for readers to understand the collection process of AirCore system.

6. In section 2.3, "coal" should be revised as methane.

Thank you, we have modified "coal" as "q (g/s) is the emission rate of $CH_4$ from coal mine"

7. What do you mean by oral ?

Sorry for this mistake, we have revised "oral" as "original" in our latest manuscript.

8. Section 2.4, This section needs to go before the model section as the location is mentioned. You should also make it clear what type of source this is as this will make a big difference to the model. Is a large open coal pit, or emission from a deep mine shaft or vent, or is it a combination of both.

Thank you for this comment, we have move this section to section 2.3 in our latest submission, and we have also indicate that the methane emission source is ventilation shaft, strong point source. We revised this sentence as follows:

"As part of the Carbon Dioxide and Methane Mission (CoMet) pre-campaign,15 active AirCore flights successfully collected data around a ventilation shaft of Pniówek coal mine on August 18, 2017 and August 21, 2017."

9.Fig2. Caption needs more explanation. What actually is the zoomed map showing? Where is the mine? Is pink the mine buidlings? is it deep mine with shaft or open pit?

Thank you for this suggestion, we are sorry for our careless to not indicate the detailed instruction of each map, and we added the information in our latest manuscript as follows:

[Figure]

"**Fig.2.** Pniówek coal mine; a. red mark represent the location of Pniówek coal mine in Poland; b. the surrounding circumstance of Pniówek coal mine, blue mark represent Pniówek coal mine; c. detailed layout of Pniówek coal mine, deep mine with shaft"

10. In section 3.1, what is a coal?

Sorry for this wrong expression, we have modified "coal" as "strong point source" in our latest manuscript as follows:

"Firstly, the dispersion of CH$_4$ emission from a strong point source is simulated by equation 1"

11. As for fig 3. Showing simulated data without showing any real 3-D data from the AirCore. What does this look like? What are you trying to simulated?

Dear reviewer, thank you for this suggestion, this suggestion is very helpful and we have revised this Fig in our latest submission as follows:

Firstly, we simulated the diffusion of CH$_4$ from strong point source by formula 1 according to the parameters in Table.R1 as follows:

Table. R1. Parameters setting of CH$_4$ diffusion

| Parameters | Actual |
|---|---|
| Emission intensity (g/s) | 180 |
| Wind speed (m/s) | 3 |
| Wind direction (°) | 90 |
| a | 0.6 |
| B | 0.7 |
| c | 0.2 |
| d | 0.6 |
| B (ppb) | 1900 |
| Emission height (m) | 50 |
| α | 0.9 |
| Accuracy of samples | 99.5% |
| Amount of samples | 99 |

$$C(x, y, z) = \frac{q}{2\pi u\sigma_y\sigma_z} \exp(\frac{-(y)^2}{2\sigma_y^2})\left\{\exp(\frac{-(z-H)^2}{2\sigma_z^2}) + \alpha \cdot \exp(\frac{-(z+H)^2}{2\sigma_z^2})\right\} + B \qquad 1$$

Then, the simulated flight track of UAV was conducted in crossing section (300 m to strong source), in Fig 7. To make this Fig 7 easier to understand, we have revised the illustrations for this Fig 7 as followed,

[Figure]

**Fig. 7.** Simulated concentration samples collected by UAV-based Aircore system. Rectangle represents crossing section perpendicular to wind direction, 300 m to point source; Red line represents simulated flight track of UAV-based Aircore system; colored points represent the $CH_4$ concentration samples in OSSEs, totally 99.

Then, the 99 points were discussed to analysis the influence of different parameters on final retrieved emission, such wind speed, wind direction, the amount and accuracy of measurement.

11.  "As shown in Table 1, q retrieved by GA-IPPF has only 0.17% bias compared with the set values.  Emission height only has 0.3 m bias to set one."  Show which is q in Table 1.

Thank you for this suggestion, q is emission rate, we have modified this sentence as follows:

"As shown in Table 1, emission rate retrieved by GA-IPPF has only 0.17% bias compared with the set values."

12. In section 3.2, "which is defined the plane coordinates of $CH_4$ samples" , something is missing here.

This suggestion is really help readers to understand, we have enriched this sentence as follows:

"In the coordinate system established in Gaussian diffusion model, wind direction determines the X axis, which further determines the position of the gas concentration sample in the two-dimensional plane (XOY)."

13. Need to make clear earlier that q = emissions in g/s.

Thank you for this helpful suggestion, and we noted that q is emission rate (g/s) in Section 2.3.1 as follows:

"q (g/s) is the emission rate of $CH_4$ from coal mine"

15. In line 221, "Wind direction determines the spatial location of the sampling point, and wrong location information leads to distinct errors in emission estimation."  This implies that there could be big errors in q if the source is not a point source, but unequally distributed over an area, which could be the case with a farm or landfill site emission.

We totally agree with your comment, if the concerned emission source is not a point source, such as farms, cattle farms or landfills, or other irregular emission sources, generally for this kind source, mass-balance approach method is usually used to quantify their emission rate.

$$F_{(CH4)} = \iint v \sin(\alpha) \cdot (C_{(x,z)} - C_{bg}) dx dz \qquad (12)$$

Where v is the wind speed, α is the angle between wind direction and the two-dimensional plane, $C_{(x,z)}$

is the density of $CH_4$ in each grid, and $C_{bg}$ is the background of $CH_4$ in each grid.

And as you mentioned, the wind direction would accuse big error in final quantification. In order to avoid readers' misunderstanding, we declare the application of GA-IPPF is mainly focus on quantitative evaluation of point sources in the full text. For examples:

We have modified our title of this manuscript as "Retrieving $CH_4$ emission rate from coal mine ventilation shaft using UAV-based AirCore observations and the GA-IPPF model"

"In this study, we developed a genetic algorithm–interior point penalty function (GA-IPPF) model to calculate the emission rate of large point sources of $CH_4$ based on concentration samples."

"Firstly, the dispersion of $CH_4$ emission from a point source is simulated by equation 1"

16.In line 225, How do you know these are errors and not limitations of instrument precision?

Dear reviewer, the errors added in wind speed, wind direction and gas concentration are set in simulation experiments, which are aimed to present the adjustment of GA-IPPF on retrieved emission rate. Each additional error would be discussed according to the controlled variables method. The original parameters settings (in red line) are shown in our latest manuscript.

Table 4. The parameters setting in dispersion simulation and the retrieved results by GA-IPPF

| Parameters | Lower boundary | Upper boundary | Actual | Retrieved |
|---|---|---|---|---|
| Emission intensity (g/s) | 0 | 100000 | 180 | 180.2±0.02 |
| Wind speed (m/s) | 0 | 100000 | 3 | 3±0.01 |
| Wind direction (°) | 70 | 110 | 90 | 90±0.10 |
| a | 0 | 1000 | 0.6 | 0.6±0.02 |
| B | 0 | 1000 | 0.7 | 0.7±0.02 |
| c | 0 | 1000 | 0.2 | 0.2±0.01 |
| d | 0 | 1000 | 0.6 | 0.6±0.01 |
| B (ppb) | 1700 | 2500 | 1900 | 1900±2.7 |
| Emission height (m) | 0 | 150 | 50 | 49.8±1.1 |
| α | 0 | 1 | 0.9 | 0.91±0.01 |

"Actual" means the set values of parameters, and "Retrieved" means the average values of parameters retrieved by GA-IPPF model through 10 000 times of simulation.

This part dose not refers to actual experiment, so it has no directly correction with the instrument precision. It worth nothing that all the set ranges of errors in the simulations are larger than the precision of instruments, for examples, the ranges of added wind error are 0 ~2.0 m/s, the ranges of added simulated gas concentration samples are 0.5 %~5.0%

17.In line 228, "The AirCore system could acquire more than 70 $CH_4$ samples in actual feasible measurements" . Not clear what this means. Number of air Core samples that can be collected and analysed within a given time period within the same source emission plume.

Sorry for this incorrect expression,. We have modified this section as follows:

Then, the simulated flight track of UAV was conducted in crossing section (300 m to strong source), in Fig 7. The spatial resolution of the supposed samples is 10 m, and 99 samples were selected from the simulated dispersion to represent the data acquired by the UAV-based AirCore.

[Figure]

**Fig. 7.** Rectangle represents crossing section perpendicular to wind direction, 300 m to point source; Red line represents simulated flight track of UAV-based Aircore system; colored points represent the $CH_4$ concentration samples in OSSEs, totally 99.

We want to model the colored point to represent the concentration samples in actual Flight. Aircore system is actual sampling continually during the Flight, the spatial resolution is 10 m in this section, which is regarded as the integral period of each value of $CH_4$ concentration in different locations.

we have modified this sentence as

"UAV-based AirCore system could acquire more than 99 $CH_4$ samples (with accuracy better than 99.5 %) in single Flight."

17.In line 240, "were "should be modified as "are"

Thank you, we revised "were" as "are" in latest manuscript.

"In this section, the performance of the GA-IPPF model and the influence of the four key input parameters are discussed."

18.In line 246, "Flight 6" says Flight 8 in text above.

Sorry for this mistake, we have modified it as Flight 6 in our latest manuscript.

19.In line 247, "spirally", in a spiral pattern

We changed "spirally" to "in a spiral pattern" in our latest submission.

20. In line 249, "samples", The Picarro measurement rate doesn't change so why 376 measurements in 7 minutes and 400 in 9 minutes?

Sorry for this careless expression, the resolution of final stored time is min, we have mistaken the store time to express the total collected time, detailed responses are shown as follows:

Actually, for Flight 6 and Flight 15, Aircore system could get about 50 samples per minute. In Flight 15, it only collected 17 samples in 2017/8/21 10:59:00 (start time). The total collected time is 8 min 04 s. similar to Flight 8, and similarly, the total collected time is 7 min 33 s in Flight 8.

| | | | | | | |
|---|---|---|---|---|---|---|
| 2 | 2017/8/21 10:59 | 18.73753 | 49.97458 | 10.38571 | 2.897267 | 3170.951 |
| 3 | 2017/8/21 10:59 | 18.73752 | 49.97459 | 10.38571 | 3.058225 | 3346.012 |
| 4 | 2017/8/21 10:59 | 18.73749 | 49.9746 | 10.38571 | 2.651128 | 3555.731 |
| 5 | 2017/8/21 10:59 | 18.73745 | 49.97463 | 10.38571 | 2.674649 | 3695.523 |
| 6 | 2017/8/21 10:59 | 18.73743 | 49.97464 | 10.38571 | 2.879437 | 3835.335 |
| 7 | 2017/8/21 10:59 | 18.73742 | 49.97466 | 10.38571 | 3.214906 | 4031.251 |
| 8 | 2017/8/21 10:59 | 18.7374 | 49.97467 | 10.38571 | 3.700835 | 4227.167 |
| 9 | 2017/8/21 10:59 | 18.73738 | 49.9747 | 10.38571 | 4.989239 | 4422.981 |
| 10 | 2017/8/21 10:59 | 18.73737 | 49.97471 | 10.38571 | 5.881123 | 4748.13 |
| 11 | 2017/8/21 10:59 | 18.73737 | 49.97473 | 10.38571 | 6.856142 | 5135.842 |
| 12 | 2017/8/21 10:59 | 18.73737 | 49.97476 | 10.38571 | 10.16905 | 5394.239 |
| 13 | 2017/8/21 10:59 | 18.73739 | 49.97481 | 10.38571 | 15.10969 | 5802.599 |
| 14 | 2017/8/21 10:59 | 18.7374 | 49.97484 | 10.38571 | 17.88322 | 6252.263 |
| 15 | 2017/8/21 10:59 | 18.73743 | 49.97489 | 10.38571 | 23.53596 | 6552.136 |
| 16 | 2017/8/21 10:59 | 18.73744 | 49.97491 | 10.38571 | 26.3497 | 6851.917 |
| 17 | 2017/8/21 10:59 | 18.73745 | 49.97494 | 10.38571 | 28.84725 | 7161.872 |
| 18 | 2017/8/21 10:59 | 18.73745 | 49.97496 | 10.38571 | 31.46561 | 7471.827 |
| 19 | 2017/8/21 11:00 | 18.73747 | 49.97504 | 10.38571 | 38.81091 | 7781.77 |
| 20 | 2017/8/21 11:00 | 18.73748 | 49.97507 | 10.38571 | 42.14035 | 8091.725 |
| 21 | 2017/8/21 11:00 | 18.7375 | 49.9751 | 10.38571 | 45.80456 | 8322.311 |
| 22 | 2017/8/21 11:00 | 18.73752 | 49.97512 | 10.33571 | 48.43182 | 8552.896 |
| 23 | 2017/8/21 11:00 | 18.73755 | 49.97517 | 10.38571 | 53.73717 | 8898.847 |

Start time in Flight 15

21. In line 254. Surely an AirCore is one sample that has enough air to allow 400 measurements when attached to the Picarro?

The AirCore is able to retrieve concentrations along the flight track because molecular diffusion inside of the AirCore is slow enough so that concentration profiles, instead of an integrated concentration average, can be retrieved. However, there is certainly mixing, especially during air sample analysis in the cavity of the CRDS analyzer, which determines the spatial resolution of the AirCore measurements. The AirCore sample was analyzed by a CRDS analyzer in a small flow rate, therefore, as many as 400 measurement points can be obtained.

22. In line 261. "5.7m and 3.64 m", Why is the calculated emission height an order of magnitude higher in the model results?

Sorry for our mistake in expression of Line 261, In two Flights, the actual emission height are 58.4 m and 35.5 m as shown in Table 1 in latest manuscript. In earlier calculation of emission rate, we ignored the actual height of ventilation shafts in Pniówek coal mine, which makes the retrieved emission height is unreasonable (5.7m and 3.64 m). We set the lower boundary of emission height is 5 m, the emission height are 58.4 m and 35.5 m respectively. We are sorry for not update the results in word expression previously. We have modified this error in our latest manuscript as follows:

"The exhaust gases of coal mine are emitted through the stack with effective emission heights of 58.4 and 35.5 m, respectively."

23. In table 2, Oral should be modified.

Thank you for this suggestion, we have modified "oral" as "original" in Table latest manuscript.

24.pay attention to superscript of expression.

Thank you, we have modified all superscript of the expressions in our latest manuscript, such as

"$R^2$", "$CH_4$".

25.In line 281, "adjust more weights" , Do you mean 'assign more weighting'

We are very grateful to you for pointing this error, "assign" is more suitable to express the actual meaning of this sentence. And we have modified "adjust" to "assign" in our latest manuscript.

26. In line 301, "a two-dimensional plane is selected according to the amount of $CH_4$ samples" should be modified.

Sorry for our wrong expression, we changed this sentence in the latest manuscript as:

"a two-dimensional plane is selected according to the flight trajectory of UAV"

27.In line 313. Added the missing date.

We have added the missing date as "Andersen et al. 2021"

28. In line 327. Why does this reference have no date and a number next to it (wrong style)

Thank you for this suggestion, we have modified all the references throughout in our latest manuscript.

29. In line 364. This is not a sentence. "To explore the reason that the acceptable difference of calculated methane emission rate by the two sources of meteorological data"

Thank you for this suggestion, we have modified this sentence as follows:

"We also explore the reason that little difference of the calculated emission rates by the two different sources of meteorological data"

30. In line 366. "were" should be modified as "are"

Thank you, we have modified "were" as "are" in latest manuscript.

31. oral should be modified

Thank you for this suggestion, we have modified "oral" as "original" in latest manuscript.

32. In line 392. Pipes leak, this is a fugitive vent emission

we are sorry for our wrong expression, and we have modified it as "the distribution of emitted gas"

33. In line 397. "lack no" this is a double negative.

Sorry for this mistake, and we have delete "no" in latest submission.

34.In section of Conclusion, Should include a recommendation of how many Air Core retrievals from a single point source are required to mininmise the errors on model results.

In this study, we focus on methane emissions from ventilation shaft in coal mine, which is treated as strong emission point source. The volatility of its emissions is significant. We assume that the intensity of methane emission is constant during the Aircore system collection. However, if methane emission rate is always fixed and a constant value, then multiple Aircore Flight would certainly improve the accuracy of quantified emission rate. Actually, methane emission rate from coal mine is always different between two Flights periods, so it is unreasonable to use multiple Flights to reduce emission error. We understand that you want us to present the data requirements of the Aircore system for the actual users, and we summarized the performance of GA-IPPF based on a certain number of samples. We recommended that the amount of samples in a single UAV-based Aircore system (with accuracy better than 99 .5%) larger than 90, the accuracy of retrieved emission rate would be better than 99 %, when error in wind speed is $\pm 0.3$ m/s and error in wind direction is $\pm 30°$.

35. As for references, need complete references, add journal volume numbers and pages / DOI where missing.

Dear reviewer, thank you for this helpful suggestion, we have added the information of journal volume numbers, pages, DOI and published year in our latest submission, for example:

*"Schneising, O., Buchwitz, M., Reuter, M., Vanselow, S., Bovensmann, H., and Burrows, J. P.: Remote sensing of methane leakage from natural gas and petroleum systems revisited, Atmos. Chem. Phys., 20, 9169–9182, https://doi.org/10.5194/acp-20-9169-2020, 2020."*

---

## Referee Report (RR1)

[referee-annotated manuscript omitted]

---

## Author Response (AR2)

Dear Reviewer

we are appreciated that your kindly and careful check for our expression in our manuscript, we have modified our manuscript as you recommended in our latest submission. And thank you again for all your helpful suggestions in the review process of this work.